# SIMPLICIAL REPRESENTATION LEARNING WITH NEURAL $k$-FORMS

**Kelly Maggs**[1]**, Celia Hacker**[2]**, Bastian Rieck**[3,4]

[1]École Polytechnique Fédérale de Lausanne (EPFL)
[2]Max Planck Institute for Mathematics in the Sciences
[3]AIDOS Lab, Institute of AI for Health, Helmholtz Munich
[4]Technical University of Munich (TUM)

## ABSTRACT

*Geometric deep learning* extends deep learning to incorporate information about the geometry and topology data, especially in complex domains like graphs. Despite the popularity of message passing in this field, it has limitations such as the need for graph rewiring, ambiguity in interpreting data, and over-smoothing. In this paper, we take a different approach, focusing on leveraging geometric information from simplicial complexes embedded in $\mathbb{R}^n$ using node coordinates. We use differential $k$-forms in $\mathbb{R}^n$ to create representations of simplices, offering interpretability and geometric consistency without message passing. This approach also enables us to apply differential geometry tools and achieve universal approximation. Our method is efficient, versatile, and applicable to various input complexes, including graphs, simplicial complexes, and cell complexes. It outperforms existing message passing neural networks in harnessing information from geometrical graphs with node features serving as coordinates.

## 1 INTRODUCTION

*Geometric deep learning* (Bronstein et al., 2017) expanded the scope of deep learning methods to include information about the geometry–and, to a lesser extent, topology—of data, thus enabling their use in more complicated and richer domains like graphs. While the recent years have seen the development of a plethora of methods, the predominant paradigm of the field remains *message passing* (Veličković, 2023), which was recently extended to handle higher-order domains, including *simplicial complexes* (Ebli et al., 2020), *cell complexes* (Hajij et al., 2020), and *hypergraphs* (Heydari & Livi, 2022). However, despite its utility, the message passing paradigm suffers from inherent limitations like over-smoothing, over-squashing, and an inability to capture long-range dependencies. These limitations often require strategies like graph rewiring, which change the underlying graph structure (Gasteiger et al., 2019; Topping et al., 2022) and thus affect generalisation performance.

Our paper pursues a completely different path and strives to leverage additional geometric information from a data set to obtain robust and interpretable representations of the input data. Specifically, we consider input data in the form of simplicial complexes embedded in $\mathbb{R}^n$ via node coordinates. This type of complex can be built from any graph with node features, with node features acting as the coordinates, for example. Our key insight is the use of *differential $k$-forms* in $\mathbb{R}^n$. A $k$-form in $\mathbb{R}^n$ can be integrated over any $k$-simplex embedded in $\mathbb{R}^n$ to produce a real number. Thus an $\ell$-tuple of $k$-forms produces an $\ell$-dimensional *representation* of the simplex independently of any message passing. From this perspective, $k$-forms play the role of globally consistent feature maps over the space of embedded $k$-simplices, possessing the geometric semantics and interpretability of integration. This enables us to use tools from differential geometry to prove a version of universal approximation, as well as a number of other theoretical results. Moreover, the structure of differential forms in $\mathbb{R}^n$ makes learning algorithms (computationally) feasible. In particular, a multi-layer perceptron with the right dimensions induces a $k$-form on $\mathbb{R}^n$ that can be integrated. This implies that *learnable*, finitely-parametrised differential forms can be woven into existing machine learning libraries and applied to common tasks in geometric deep learning. We find that our method is better capable

of harnessing information from geometrical graphs than existing message passing neural networks. Next to being *efficient*, our method is also *generally applicable* to a wide variety of input complexes, including graphs, simplicial complexes, and cell complexes.[1]

> **In a nutshell:** We consider DATA in the form of simplicial chains on *embedded simplicial complexes*, defining learnable *differential k-forms* as FEATURE MAPS, and introduce the concept of an *integration matrix*, which serves as an overall REPRESENTATION of the data.

**Organisation of Paper.** We present the relevant background for simplicial complexes and differential forms in Section 2. In Section 3, we introduce neural $k$-forms and prove a universal approximation statement. We also show how neural $k$-forms induce a so-called integration matrix, and use the properties of integration to prove a number of propositions. In Section 4 we present the basic architecture and algorithms. Finally, in Section 5, we perform several intuitive experiments and benchmark our method on standard geometrical deep learning data sets.

**Related Work.** Several methods focus on generalising graph neural networks (GNNs) to higher-dimensional domains, proposing *simplicial neural networks* (Bodnar et al., 2021b; Bunch et al., 2020; Ebli et al., 2020; Giusti et al., 2022; Goh et al., 2022; Keros et al., 2022; Roddenberry et al., 2021), methods that can leverage higher-order topological features of data (Hajij et al., 2023; Hensel et al., 2021; Horn et al., 2022), or optimisation algorithms for learning representations of simplicial complexes (Hacker, 2020). All of these methods operate on simplicial complexes via *diffusion processes* or *message passing* algorithms. Some works also extend message passing or aggregation schemes to *cell complexes* (Bodnar et al., 2021a; Hajij et al., 2020; 2023) or *cellular sheaves* (Barbero et al., 2022; Hansen & Gebhart, 2019). However, these existing methods exhibit limitations arising from the use of message passing or aggregation along a combinatorial structure. Message passing often results in *over-smoothing* (a regression to the mean for all node features, making them indistinguishable) or *over-squashing* (an inability to transport long-range node information throughout the graph), necessitating additional interventions (Gasteiger et al., 2019; Nguyen et al., 2023; Topping et al., 2022). Hence, there is a need for methods that go beyond message passing. Our work provides a novel perspective based on the integration of learnable $k$-forms on combinatorial domains like graphs or simplicial complexes embedded in $\mathbb{R}^n$, i.e. we assume the existence of (vertex) coordinates.

## 2 BACKGROUND

This section introduces the required background of simplicial complexes and differential forms. We restrict our focus to simplices and differential forms in $\mathbb{R}^n$, given this is the only setting we will use in practice to make the theory more accessible. For additional background references, we recommend Nanda (2021) for computational topology and Lee (2003) or Tao (2009) for differential forms. More details are also provided in Appendix A.

**Abstract Simplicial Complexes.** An abstract simplicial complex $\mathcal{S}$ is generalisation of a graph on a set of vertices $\mathcal{S}^0$. In a graph, we have pairwise connections between nodes given by edges, or 1-dimensional simplices (denoted by $\mathcal{S}^1$). In simplicial complexes, there are higher-dimensional counterparts of these connections, called the $k$-dimensional simplices, or just $k$-simplices. The $k$-simplices are connections between $k + 1$ vertices of the set $\mathcal{S}^0$, with a 2-simplex forming a triangle, a 3-simplex forming a tetrahedron, and so on. We denote a $k$-simplex by $\sigma = [v_0, \ldots, v_k]$, where $v_i \in \mathcal{S}^0$, writing $\mathcal{S}^k$ to refer to the set of all $k$-simplices. If a simplex $\sigma \in \mathcal{S}$, then so are all of its faces, i.e. all simplices formed by subsets of the vertices of $\sigma$.

**Affine Embeddings.** Data in geometric deep learning most often comes as a geometric object—a graph or simplicial complex—combined with node features in $\mathbb{R}^n$. Formally, node features correspond to a *node embedding* of a simplicial complex $\mathcal{S}$, which is a map $\phi\colon \mathcal{S}^0 \to \mathbb{R}^n$. The *standard geometric k-simplex* is the convex hull $\Delta_k = [0, t_1, \cdots, t_k] \subset \mathbb{R}^k$, where $t_i$ is the endpoint of the $i$-th basis vector. For each $k$-simplex $\sigma = [v_0, \ldots, v_k]$ the map $\phi$ induces an *affine embedding* $\phi_\sigma\colon \Delta^k \to \mathbb{R}^n$ whose image is the convex hull $[\phi(v_0), \ldots, \phi(v_k)]$.

---

[1]For example, by taking barycentric subdivision, integration of forms over cell complexes is recoverable by integration over simplicial complexes.

**Chains and Cochains over $\mathbb{R}$.** For an oriented[2] simplicial complex $\mathcal{S}$, the *simplicial k-chains* $C_k(\mathcal{S}; \mathbb{R})$ are the vector space

$$C_k(\mathcal{S}; \mathbb{R}) = \left\{ \sum_i \lambda_i \sigma_i \mid \sigma_i \in \mathcal{S}^k, \lambda_i \in \mathbb{R} \right\} \tag{1}$$

of formal linear combinations of $k$-simplices in $\mathcal{S}$. The *simplicial k-cochains* $C^k(\mathcal{S}; \mathbb{R})$ over $\mathbb{R}$ are the dual space $\mathsf{Hom}(C_k(\mathcal{S}); \mathbb{R})$ of linear functionals over the simplicial $k$-chains.

**Differential Forms.** The *tangent space* $T_p(\mathbb{R}^n)$ at $p$ is the space of all vectors originating at a point $p \in \mathbb{R}^n$, and its elements are called *tangent vectors*. In our case, this is space is canonically isomorphic to the underlying space $\mathbb{R}^n$. A differential form is a function that assigns a notion of *volume* to tuples of tangent vectors at each point in $\mathbb{R}^n$. Given a tuple of $k$ standard basis vectors $(e_{i_1}, \cdots, e_{i_k})$ in $\mathbb{R}^n$ indexed by $I = (i_1, i_2, \ldots, i_k)$ and a vector $v \in T_p(\mathbb{R}^n)$, the vector $v^I$ is the projection of $v$ onto the $I$-subspace spanned by $(e_{i_1}, e_{i_2}, \ldots, e_{i_k})$. The associated *monomial k-form*

$$dx_I(v_1, v_2, \ldots, v_k) = \varepsilon^I(v_1, v_2, \ldots, v_k) := \det\left[v_1^I, v_2^I, \ldots, v_k^I\right] \tag{2}$$

represents the standard volume spanned by $k$ tangent vectors $v_i \in T_p(\mathbb{R}^n)$ in the $I$-subspace of the tangent space.

**Scaling Functions.** General differential forms are built from locally linear combinations of re-scaled monomial $k$-forms. Formally, a general differential $k$-form $\omega \in \Omega^k(\mathbb{R}^n)$ can be written as

$$\omega_p(v_1, v_2, \ldots, v_k) = \sum_I \alpha_I(p) dx_I(v_1, v_2, \ldots, v_k), \tag{3}$$

where $p \in \mathbb{R}^n$ and $v_i \in T_p(\mathbb{R}^n)$ and $I$ ranges over the $\binom{n}{k}$ subspaces spanned by sets of $k$ basis vectors in $\mathbb{R}^n$. The *scaling functions* $\alpha_I \colon \mathbb{R}^n \to \mathbb{R}$ represent a re-scaling of the standard volume in the $I$-subspace for each point $p \in \mathbb{R}^n$. Intuitively, the scaling functions in a differential $k$-form specify the size and orientation of a subspace for each point.

**Integration.** Differential $k$-forms can be integrated over embedded $k$-simplices $\phi \colon \Delta^k \to \mathbb{R}^n$. Let $D\phi(t)$ be the Jacobian matrix of $\phi$ at $t \in \Delta^k$. For affinely-embedded simplices, $D\phi(t)$ is given by

$$D\phi_{i,j} = \left[\phi^i(t_j) - \phi^i(0)\right]_{i,j}. \tag{4}$$

The integral of $\omega \in \Omega^k(\mathbb{R}^n)$ over the image of $\phi$ can be expressed as

$$\int_\phi \omega := \sum_I \int_{\Delta^k} \alpha_I(\phi(t))\varepsilon^I(D\phi)dt, \tag{5}$$

where the integral is interpreted as a standard Riemann integral over the (compact) subset $\Delta^k \subseteq \mathbb{R}^k$. This represents the signed volume of the image of $\phi$ with respect to the differential form $\omega$. In practice, this integral is approximated with a finite Riemann sum. We provide more background on integration of forms in Appendix A. The integral in Eq. (5) is well-defined if $\phi$ is a $C^1$ embedding function and $\alpha_I \circ \phi$ is integrable.

## 3 NEURAL $k$-FORMS AND INTEGRATION MATRICES

Broadly speaking, representation learning is the process of turning data into vectors in $\mathbb{R}^\ell$, followed by the use of standard tools of machine learning to classify/predict attributes based on these vector representations. In graph learning and simplicial learning more generally, data comes in the form of simplicial complexes embedded in $\mathbb{R}^n$. One simple vectorization is to take the standard *volume* of a $k$-simplex embedded in $\mathbb{R}^n$. **The central idea of our paper is to create vector representations of $k$-simplices as volumes relative to a tuple of differential $k$-forms.** Indeed, a tuple of $k$-forms $\omega_1, \omega_2, \ldots, \omega_\ell \in \Omega^k(\mathbb{R}^n)$ determines a representation function

$$(\phi \colon \Delta^k \to \mathbb{R}^n) \mapsto \left( \int_\phi \omega_1, \ldots, \int_\phi \omega_\ell \right) \in \mathbb{R}^\ell \tag{6}$$

---

[2]The choice of orientation of each simplex corresponds to a choice of sign for each basis vector.

that vectorises any $k$-simplex embedded in $\mathbb{R}^n$ by calculating its volume via integration of forms. In this paradigm, each $k$-form takes the role of a *feature map* on the space of embedded $k$-simplices.

**Neural $k$-forms.** The key to making integral-based representations *learnable* is to model the scaling functions in Eq. (3) as a multi-layer perceptron (MLP). For a set of $k$ basis vectors $I$, let $\pi_I \colon \mathbb{R}^{\binom{n}{k}} \to \mathbb{R}$ be the *projection onto the $I$-subspace* and define $\psi_I := \pi_I \circ \psi \colon \mathbb{R}^n \to \mathbb{R}$.

**Definition 1.** Let $\psi \colon \mathbb{R}^n \to \mathbb{R}^{\binom{n}{k}}$ be an MLP. The *neural $k$-form* $\omega^\psi \in \Omega^k(\mathbb{R}^n)$ associated to $\psi$ is the $k$-form $\omega^\psi = \sum_I \psi_I dx_I$.

In words, the components of the MLP correspond to the scaling functions of the neural $k$-form. **The goal of a neural $k$-form is thus to learn the size and orientation of $k$-dimensional subspaces at each point in $\mathbb{R}^n$ according to some downstream learning task.** In case the activation function is a sigmoid function or $\mathtt{tanh}$, Definition 1 produces smooth $k$-forms, whereas for a $\mathtt{ReLU}$ activation function, one obtains piecewise linear $k$-forms.

**Remark 2.** A neural 0-form over $\mathbb{R}^n$ with $\ell$ features is an MLP $\psi$ from $\mathbb{R}^n$ to $\mathbb{R}^\ell$. The 0-simplices $p \in \mathbb{R}^n$ are points in $\mathbb{R}^n$, and integration of a 0-form corresponds to the evaluation $\psi(p)$. In this way, neural $k$-forms are a direct extension of MLPs to higher-dimensional simplices in $\mathbb{R}^n$.

**Universal Approximation.** We would like to know which $k$-forms on $\mathbb{R}^n$ are possible to approximate with neural $k$-forms. The following proposition translates the well-known Universal Approximation Theorem (Cybenko, 1989; Hornik et al., 1989) for neural networks into the language of neural $k$-forms. Here, the norm $\| \ \|_{\Omega_c(\mathbb{R}^n)}$ on $k$-forms is induced by the standard Riemannian structure on $\mathbb{R}^n$, as explained in Appendix C.

**Theorem 3.** *Let $\alpha \in C(\mathbb{R}, \mathbb{R})$ be a non-polynomial activation function. For every $n \in \mathbb{N}$ and compactly supported $k$-form $\eta \in \Omega_c^k(\mathbb{R}^n)$ and $\epsilon > 0$ there exists a neural $k$-form $\omega^\psi$ with one hidden layer such that $\|\omega^\psi - \eta\|_{\Omega_c(\mathbb{R}^n)} < \epsilon$.*

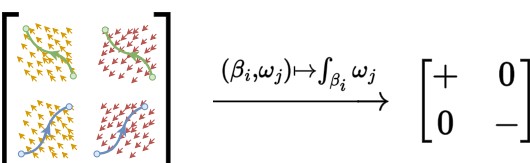

Figure 1: An *integration matrix* with data in dimension 1, where embedded oriented 1-simplices correspond to paths and 1-forms are canonically identified with vector fields (see Appendix A for details). Integration of a 1-form against a path corresponds to path integration against the respective vector field. Thus, integration of the paired paths and 1-forms in the left matrix recovers real values with the signs given in the right matrix.

**Cochain Matrices.** Most simplicial neural networks follow a similar procedure. The key step is that the $k$-simplices of each simplicial complex $\mathcal{S}$ are replaced by a matrix $X_{\mathcal{S}}(\beta, \gamma)$ containing a selection of $\ell$ simplicial cochains $(\gamma_1, \ldots, \gamma_\ell)$ as columns. This can be thought of as a matrix containing evaluations $\gamma_j(\beta_i)$ with respect to some basis $(\beta_1, \ldots, \beta_s)$ for the simplicial $k$-chains $C_k(\mathcal{S}; \mathbb{R})$. However, there is no canonical initialisation when one starts with only the data of a set of embedded finite simplicial complexes.[3] We address this issue by introducing the *integration matrix* induced by a neural $k$-form. This produces the same data type, but has the advantage that the feature cochains correspond to integration of the *same form* defined over the ambient space.

**Integration Matrices.** For an embedded finite simplicial complex, an $\ell$-tuple $\omega = (\omega_1, \omega_2, \ldots, \omega_\ell)$ of $k$-forms induces a matrix suitable to simplicial learning algorithms in a natural way via integration. Let $\phi \colon \mathcal{S} \to \mathbb{R}^n$ be an affine embedding of a simplicial complex and $\beta = (\beta_1, \beta_2, \ldots, \beta_m) \in C_k(\mathcal{S}; \mathbb{R})$ be a set of specified $k$-chains.

**Definition 4.** The *integration matrix* with respect to $\beta$ and $\omega$ is $X_\phi(\beta, \omega) = \left[ \int_{\beta_i} \omega_j \right]_{i,j}$.

The integral of $\omega$ over a simplicial chain $\beta = \sum \lambda_\sigma \sigma \in C_k(\mathcal{S}; \mathbb{R})$ with respect to the embedding $\phi \colon \mathcal{S} \to \mathbb{R}^n$ corresponds to the integral

$$\int_\beta \omega = \sum \lambda_\sigma \int_{\phi_\sigma} \omega. \tag{7}$$

---

[3]Indeed, if one takes a random initialisation, the feature cochains do not have a shared interpretable meaning across different complexes.

**Remark 5** (Interpretability). The key point is that the integration matrices of two different simplicial complexes $\phi\colon \mathcal{S} \to \mathbb{R}^n$ and $\phi'\colon \mathcal{S}' \to \mathbb{R}^n$ embedded in $\mathbb{R}^n$ have a shared interpretation. That is, the $j$-th column of both of their integration matrices corresponds to the volume of $k$-simplices against *the same feature $k$-form $\omega_j \in \Omega^k(\mathbb{R}^n)$.*

A tuple of $\ell$ neural networks $\psi_j\colon \mathbb{R}^n \to \mathbb{R}^{\binom{n}{k}}$ with associated neural $k$-forms $\omega^{\psi_j}$ induces a *learnable* matrix representation

$$(\beta, \psi) \mapsto X_\phi(\beta, \omega^\psi) = \Big[ \int_{\beta_i} \omega^{\psi_j} \Big] \tag{8}$$

of the given simplicial chain data $\beta$ with embedding $\phi\colon \mathcal{S} \to \mathbb{R}^n$. This intermediate representation is finitely parametrised by $\psi$ and can thus be updated via backpropagation. The matrix representation of a simplex—and by extension a simplicial chain—depends on whether the neural $k$-form decides that it is embedded in a large or small subspace, and with what orientation.

**Basic Properties.** There are a number of useful basic properties about integration matrices that follow from the well-known properties of integration. In the next proposition, we conceptualise

$$\beta = (\beta_1, \dots, \beta_m)^T \in M^{m \times 1}\Big(C_k(\mathcal{S}; \mathbb{R})\Big) \text{ and } \omega = (\omega_1, \dots, \omega_\ell) \in M^{1 \times \ell}\Big(\Omega^k(\mathbb{R}^n)\Big) \tag{9}$$

as chain-valued column and $k$-form valued row vectors, respectively. Real-valued matrices act on both vectors by scalar multiplication and linearity.

**Proposition 6** (Multi-linearity). *Let $\phi\colon \mathcal{S} \to \mathbb{R}^n$ be an embedded simplicial complex. For any matrices $L \in M^{m' \times m}(\mathbb{R})$ and $R \in M^{\ell \times \ell'}(\mathbb{R})$, we have*

$$X_\phi(L\beta, \omega R) = L X_\phi(\beta, \omega) R \tag{10}$$

A staple requirement of geometric deep learning architectures is that they should be permutation and orientation equivariant. In our setting, these properties are a direct corollary of Proposition 6.

**Corollary 7** (Equivariance). *Let $\beta = (\beta_1, \dots, \beta_m)$ be a basis for the $k$-chains $C_k(\mathcal{S}; \mathbb{R})$ of an embedded oriented simplicial complex $\phi\colon \mathcal{S} \to \mathbb{R}^n$.*

1. *(Permutation) $X_\phi(P\beta, \omega) = P X_\phi(\beta, \omega)$ for all permutation matrices $P \in M^{m \times m}(\mathbb{R})$.*

2. *(Orientation) $X_\phi(Q\beta, \omega) = Q X_\phi(\beta, \omega)$ for all signature matrices[4] $Q \in M^{m \times m}(\mathbb{R})$.*

## 4 ARCHITECTURE

**Embedded Chain Data.** The input data $\mathcal{D} = \{(\mathcal{S}_\alpha, \phi_\alpha, \beta_\alpha)\}$ to our learning pipeline consists of a set of triples $(\mathcal{S}_\alpha, \phi_\alpha, \beta_\alpha)$ of *embedded chain data*, where

- $\mathcal{S}_\alpha$ is a simplicial complex
- $\phi_\alpha\colon \mathcal{S}_\alpha^0 \to \mathbb{R}^n$ is an affine embedding and
- $\beta_\alpha \in \bigoplus_{m_\alpha} C_k(\mathcal{S}; \mathbb{R})$ is a tuple of $m_\alpha$ data $k$-chains on $\mathcal{S}_\alpha$.

If no chains are provided, one can take the standard basis of oriented $k$-simplices of each simplicial complex as the input chains. The canonical example is a dataset consisting of graphs $\{(\mathcal{G}_\alpha, \phi_\alpha, \beta_\alpha)\}$ with node features $\phi_\alpha\colon \mathcal{G}_\alpha^0 \to \mathbb{R}^n$ and $\beta_\alpha$ corresponding to the standard edge chains with arbitrary orientations.

**Approximating Integration Matrices.** The main departure from standard geometric deep learning architectures is the transformation from embedded $k$-chain data to integration matrices. The high-level structure of this process is presented in Algorithm 1. Given a tuple of neural $k$-forms, each represented by an MLP $\psi_j\colon \mathbb{R}^n \to \mathbb{R}^{\binom{n}{k}}$, integrals of embedded $k$-simplices $\phi_\sigma\colon \Delta^k \to \mathbb{R}^n$ are calculated by a finite approximation (Appendix B) of the integral formula

$$\int_{\phi_\sigma} \omega_j = \sum_I \int_{\Delta^k} \psi_{I,j}(\phi_\sigma(t)) \varepsilon^I\big(D\phi_\sigma(t)\big) dt \approx \mathtt{VolApprox}\Big(\int_{\phi_\sigma} \omega_j\Big) \tag{11}$$

---

[4]A signature matrix is a diagonal matrix with $\pm 1$ entries.

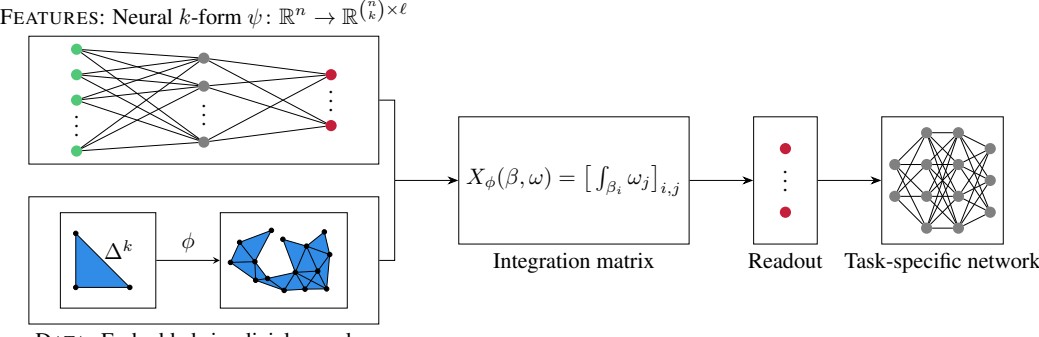

FEATURES: Neural $k$-form $\psi\colon \mathbb{R}^n \to \mathbb{R}^{\binom{n}{k} \times \ell}$

Integration matrix

Readout

Task-specific network

DATA: Embedded simplicial complex

Figure 2: A schematic of our proposed neural $k$-form learning architecture.

---

**Algorithm 1** Generate Integration Matrix

---

**Inputs:** Embedded chain data $(\mathcal{S}, \phi, \beta)$ with $k$-simplices $\sigma \in \mathcal{S}^k$;
$k$-chains $\beta_i = \sum \lambda_{\sigma,i}\sigma \in C_k(\mathcal{S}; \mathbb{R})$; $i = 1, \dots, m$;
MLPs[5] $\psi_j\colon \mathbb{R}^n \to \mathbb{R}^{\binom{n}{k}}$; $j = 1, \dots, \ell$.

---

$X = \mathbf{0} \in M_{m,\ell}(\mathbb{R})$  ▷ Initialise $X$ as the 0-matrix with $m \times \ell$ elements
**for** $1 \le j \le \ell$ **do**  ▷ Iterate over forms
    **for** $1 \le i \le m$ **do**  ▷ Iterate over chains
        $X_{i,j} \leftarrow \sum \lambda_{\sigma,i}\texttt{VolApprox}(\int_\sigma \omega^{\psi_j})$
**Return:** $X_\phi(\beta, \omega^\psi) = X$  ▷ Return Integration Matrix

---

appearing in Eq. (5). Integrals of chains $\beta_i$ are calculated as linear combinations. **The entries of the integration matrix $X_\phi(\beta, \omega^\psi)$ thus depend in a differentiable manner on the component functions $\psi_{I,j}$ of the underlying MLP.**

**Readout Layers.** Once an embedded simplicial complex is transformed into an integration matrix, it is then fed into a *readout layer* (as in the case for standard simplicial or graph neural networks). The output of a readout layer is a single representation of the entire complex, and should not depend on the number of simplices if one wishes to compare representations among different complexes. Common read-out layers include summing column entries, and $L_1$ or $L_2$ norms of the columns. Note that only the latter two are invariant under change of orientation.

**Neural $k$-form Backpropagation.** For a fixed dataset of embedded $k$-chains, neural $k$-forms are the learnable feature functions. Algorithm 2 and Figure 2 illustrate the basic pipeline for performing backpropagation of neural $k$-forms over embedded chain data and a loss function. Neural $k$-forms and a user-determined classifier are randomly initialised using *any* standard MLP initialisation. Each embedded chain data $(\mathcal{S}_\alpha, \phi_\alpha, \beta_\alpha)$ is transformed into an integration matrix by Algorithm 1, before being read out, classified, and evaluated against a loss function.

**Implementation.** Our methods can be realised using any deep learning framework that permits training an MLP. We created a proof-of-concept implementation using `PyTorch-Geometric` (Fey & Lenssen, 2019) and `PyTorch-Lightning` (Falcon & The PyTorch Lightning team, 2019) and make it publicly available under `https://github.com/aidos-lab/neural-k-forms`.

## 5 EXPERIMENTS AND EXAMPLES

This section presents examples and use cases of neural $k$-forms in classification tasks, highlighting their *interpretability* and *computational efficiency*.

---

**Algorithm 2** Neural $k$-form Backpropagation

---

**Data:** $\mathcal{D} = \{(\mathcal{S}_\alpha, \phi_\alpha, \beta_\alpha) \mid \phi_\alpha \colon \mathcal{S}_\alpha \to \mathbb{R}^n, \beta_\alpha \in \bigoplus_{m_\alpha} C_k(\mathcal{S}; \mathbb{R}), m_\alpha \in \mathbb{N}\}$

**Initialise** MLPs $\psi_j \colon \mathbb{R}^n \to \mathbb{R}^{\binom{n}{k}}; 1 \leq j \leq \ell$; classifier $\eta \colon \mathbb{R}^\ell \to \mathbb{R}^{\ell'}$.

---

**for** $(\mathcal{S}_\alpha, \phi_\alpha, \beta_\alpha) \in \mathcal{D}$ **do**
    $X_{\phi_\alpha}(\beta_\alpha, \omega^\psi) \leftarrow \texttt{IntegrationMatrix}(\phi_\alpha, \beta_\alpha, \psi)$
    $X \leftarrow \texttt{Readout}(X_{\phi_\alpha}(\beta_\alpha, \omega^\psi))$           ▷ Vector Representation
    $X \leftarrow \eta(X)$           ▷ Prediction
    $L = \texttt{Loss}(X)$
    $\texttt{Backward}(L, \psi, \eta)$           ▷ Update $k$-forms and Classifier

---

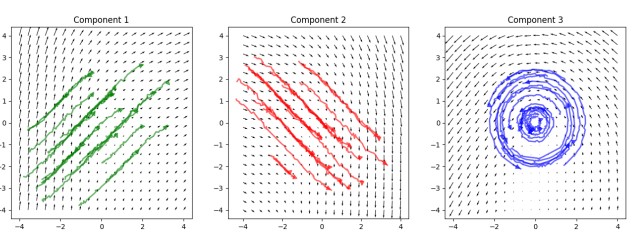 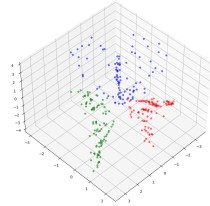

(a) Learned 1-forms corresponding to paths in each class.      (b) Path representations.

Figure 3: Synthetic Path Classification via Learnable 1-forms. The 1-form (a vector field in this case) adjusts itself to the data, resulting in distinct path representations.

## 5.1 SYNTHETIC PATH CLASSIFICATION

Our first experiment is classifying paths in $\mathbb{R}^2$. The goal of the experiment is pedagogical; it illustrates how to interpret the learned 1-forms rather than benchmark the method. A piecewise linear path in $\mathbb{R}^n$ is a simplicial complex determined by an ordered sequence of node embeddings. The 1-simplices are linear maps $\sigma_i \colon I \to \mathbb{R}^n$ from the $i$-th embedded node and to the $(i+1)$-st embedded node, where the full path corresponds to the 1-chain $\sum_i \sigma_i$. The integral of this chain against a 1-form corresponds to the path integral. Figure 3a shows three classes of piecewise linear paths[6] in $\mathbb{R}^2$ that we will classify using our method. The idea is that we will learn three 1-forms, which correspond to each class. To initialise the three 1-forms in $\mathbb{R}^2$, we randomly initialise an MLP $\psi \colon \mathbb{R}^2 \to \mathbb{R}^{2 \times 3}$. A forward pass consists of two stages: *integration* of the 1-simplices in the path against each of the three forms to produce an integration matrix, followed by taking a *column sum* and applying $\texttt{softmax}$ to produce a distribution over which we perform $\texttt{CrossEntropyLoss}$. The $i$-th column sum corresponds to a path integral of the path against the $i$-th form; the prediction of a path is thus determined by which feature 1-form produces the highest path integral. Backpropagation against this loss function thus attempts to modify the $i$-th 1-form so that it produces a more positive path integral against the paths in class $i$ and more negative otherwise.

Figure 3a also shows the feature 1-forms as vector fields over their corresponding classes of paths. Note that the vector fields are randomly initialised and updated while the paths are the fixed data points. The learned 1-forms of each class resemble vector fields that roughly reproduce the paths in their class as integral flow lines, and are locally orthogonal or negatively correlated to paths in the other classes. This is a direct result of the objective function, which attempts to maximise the path integral within each class and minimise it for others. Figure 3b depicts the paths as points coloured by class, where the coordinates correspond to the path integrals against the three learned 1-forms. We observe a clear separation between the classes, indicating that the representations trivialise the downstream classification task. We also compare the model with a standard MLP in Appendix E.

---

[6]The orientation is indicated by the arrow at the end of each path.

Table 1: Results (mean accuracy and standard deviation of a 5-fold cross-validation) on small graph benchmark datasets that exhibit 'geometrical' node features. Parameter numbers are approximate because the number of classes differ.

| | Params. | BZR | COX2 | DHFR | Letter-low | Letter-med | Letter-high |
|---|---|---|---|---|---|---|---|
| EGNN (Satorras et al., 2021) | 1M | $79.51 \pm 1.87$ | $78.16 \pm 0.46$ | $64.02 \pm 2.68$ | $93.20 \pm 0.68$ | $65.91 \pm 1.60$ | $\mathbf{68.98 \pm 1.80}$ |
| GAT (Veličković, 2022) | 5K | $\mathbf{81.48 \pm 2.90}$ | $80.73 \pm 2.45$ | $\mathbf{73.02 \pm 2.54}$ | $90.04 \pm 2.23$ | $63.69 \pm 5.97$ | $43.73 \pm 4.13$ |
| GCN (Kipf & Welling, 2017) | 5K | $79.75 \pm 0.68$ | $\mathbf{79.88 \pm 1.65}$ | $70.12 \pm 5.43$ | $81.38 \pm 1.57$ | $62.00 \pm 2.07$ | $43.06 \pm 1.67$ |
| GIN (Xu et al., 2019) | 9K | $79.26 \pm 1.03$ | $78.38 \pm 0.79$ | $68.52 \pm 7.38$ | $85.00 \pm 0.59$ | $67.07 \pm 2.47$ | $50.93 \pm 3.47$ |
| N$k$F (ours) | 4K | $78.77 \pm 0.55$ | $80.30 \pm 2.42$ | $64.42 \pm 2.09$ | $\mathbf{93.42 \pm 1.94}$ | $\mathbf{67.69 \pm 1.28}$ | $62.93 \pm 4.13$ |

## 5.2 SYNTHETIC SURFACE CLASSIFICATION

This example demonstrates how our framework is of interest to deal with higher-dimensional data, i.e. simplicial complexes of dimension $k \geq 2$. Conceptually, the difficulty is going from 1-dimensional objects to $k$-dimensional objects with any $k \geq 2$. We restrict ourselves to $k = 2$ since higher dimensions are similar to this case. The data we consider here are triangulated surfaces embedded in $\mathbb{R}^3$, the underlying combinatorial complex is always the same— a triangulation of a square—but the embeddings are different. For a given surface, the embedding of each 2-simplex is given by linear interpolation between the coordinates of its three vertices in $\mathbb{R}^3$. We consider two classes of surfaces, in the first class the embeddings of the nodes are obtained by sampling along a sinusoidal surface in the $x$-direction, while the second class is given by a surface following a sinusoidal shape in the $y$-direction, in each case with added translation and noise. We use learnable 2-forms $\omega_1, \omega_2$ on $\mathbb{R}^3$ represented by an MLP $\psi \colon \mathbb{R}^3 \to \mathbb{R}^{3 \times 2}$. In a forward pass of the model, we integrate the 2-forms given by the MLP over the 2-simplices of a surface. Each point in $\mathbb{R}^3$ has a value in $\mathbb{R}^{3 \times 2}$ given by the MLP evaluated at that point. The process of integrating over the 2-simplices corresponds to integrating the point-wise evaluation of the MLP over the regions of $\mathbb{R}^3$ defined by each embedded 2-simplex. This process gives the integration matrix of the forms $\omega_1, \omega_2$ over the 2-simplices of the complex.

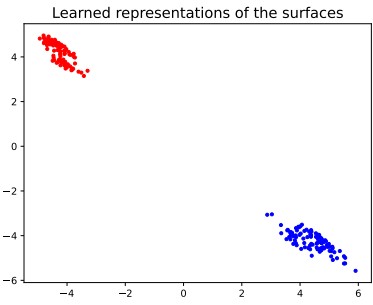

Figure 4: Synthetic surface classification via learnable 2-forms.

The next step is the readout layer, which sums the entries in each column of the integration matrix, corresponding to summing the value of each cochain over the simplicial complex, giving the total surface integral. As there are two 2-forms this yields a vector representation in $\mathbb{R}^2$, each entry corresponding to a 2-form. Finally, these representations are then passed through a `softmax` for classification and `CrossEntropyLoss` is used as a loss function. The MLP is then updated by backpropagation. Figure 4 plots the representations in $\mathbb{R}^2$ learned through the model described above. Each point represents a surface in the data set and the colour is given by the class of the corresponding surface, showing the clear separation learned by the neural 2-forms. Further details can be found in Appendix E.2.

## 5.3 REAL-WORLD GRAPHS

In this experiment we attempt to use our model to leverage the geometry of non-equivariant node features for *graph classification* on a set of benchmark datasets. The basic architecture of the model follows that described in the *Architecture and Parameters* section. Graphs are represented as a 1-chain consisting of all their edges. We randomly initialise a set of $\ell$ feature 1-forms, produce and read-out the integration matrices before feeding through a simple MLP classifier and performing backpropagation. We use an $L_2$-column readout layer so the network is invariant under edge orientations. Please refer to Appendix E.3 for specific architectural details. We use state-of-the-art graph neural networks (GNNs) following a recently-described benchmark (Dwivedi et al., 2023), experimenting with different numbers of layers. As an additional comparison partner, we also use a recent equivariant GNN architecture that is specifically geared towards analysing data with an underlying geometry (Satorras et al., 2021). Table 1 depicts the results on smaller graph datasets in the TU dataset (Morris et al., 2020). Here the node features that carry both equivariant information (corresponding to 3D coordinates of molecules, for instance) and non-equivariant information (one-hot atomic type, weight,

Table 2: Results (mean AUROC and standard deviation of 5 runs) on benchmark datasets from the 'MoleculeNet' database (Wu et al., 2018). While the GNNs also train for smaller numbers of parameters, we observed significant drops in predictive performance. We thus report only the best results for GNNs, using the most common models described in the literature.

|  | Params. | BACE | BBBP | HIV |
|---|---|---|---|---|
| GAT (Veličković et al., 2018) | 135K | $69.52 \pm 17.52$ | $76.51 \pm 3.36$ | $56.38 \pm 4.41$ |
| GCN (Kipf & Welling, 2017) | 133K | $66.79 \pm 1.56$ | $73.77 \pm 3.30$ | $68.70 \pm 1.67$ |
| GIN (Xu et al., 2019) | 282K | $42.91 \pm 18.56$ | $61.66 \pm 19.47$ | $55.28 \pm 17.49$ |
| N$k$F (ours) | 9K | $\mathbf{83.50 \pm 0.55}$ | $\mathbf{86.41 \pm 3.64}$ | $\mathbf{76.70 \pm 2.17}$ |

etc.). For the non-equivariant models (ours, GCN, GAT) the position features are omitted. Overall, our method exhibits competitive performance, in particular given the fact that it does not make use of *any* message passing and has a smaller parameter footprint. In Table 2 we compare our model on the larger datasets in the 'MoleculetNet' benchmark (Wu et al., 2018). Note that the datasets we have chosen have no provided 'positional' node features, so the given node features (i.e. atomic weights, valence, etc.) are not equivariant and cannot be compared with EGNN. As Table 2 shows, our model based on neural $k$-forms outperforms state-of-the-art graph neural networks in terms of AUROC, using a fraction of the number of parameters (this also holds for accuracy and average precision, which are, however, not typically reported for these data sets).

# 6 DISCUSSION

**Summary.** We developed *neural k-forms*, a method for learning representations of simplicial cochains to solve tasks on embedded graphs and simplicial complexes. Deviating from the predominant paradigms in *geometric deep learning*, our method adopts a fundamentally different novel perspective based on integration of forms in the ambient feature space. We have shown the feasibility of the method through a comprehensive experimental suite, demonstrating its effectiveness using only a very small number of parameters. Notably, our method does not utilise any kind of message passing, and we hypothesise that it is possible that this implies that issues like *over-smoothing* may affect our method less than graph neural networks.

**Limitations.** A conceptual limitation is that we require (at least) the existence of node or vertex coordinates, i.e. our method only operates on embedded complexes. The computational feasibility of higher order $k$-forms in large embedding spaces is another possible limitation. Indeed, the number of monomial $k$-forms in $\mathbb{R}^n$ is $\binom{n}{k}$, and similar issues arise for numerical integration over higher-dimensional simplices. Further, we have only benchmarked our method on graph classification tasks. It remains to be seen whether the method performs as well on graph regression tasks, as well as for benchmark learning tasks on higher-dimensional simplicial complexes.[7] Finally, our model currently cannot deal with node features that are equivariant with respect to the embedding space.

**Outlook.** We aim to study these issues, as well as the behaviour of our methods in the context of long-range dependencies, in a follow-up work. In addition, since our neural $k$-form formulation is equivalent to an MLP, the learning process may benefit from the plethora of existing methods and tricks that are applied to optimise MLPs in practice. We argue that our experiments point towards the utility of using a geometric interpretation of our representations as *integrals* over $k$-forms may provide valuable insights to practitioners. Lastly, we provide a small example of a *Convolutional 1-form network* in Appendix D that may lead to better incorporation of equivariant node embeddings. We provide this auxiliary example as part of a broader future work program of rebuilding common ML architectures on top of neural $k$-forms rather than message passing schemes.

---

[7]We note that a comprehensive, agreed-upon framework for benchmarking simplicial neural networks has yet to be established.

ACKNOWLEDGEMENTS

B.R. is supported by the Bavarian state government with funds from the *Hightech Agenda Bavaria*. K.M. received funding from the European Union's Horizon 2020 research and innovation programme under the Marie Skłodowska-Curie grant agreement No 859860. The authors gratefully acknowledge the Leibniz Supercomputing Centre for funding this project by providing computing time on its Linux cluster. The authors also acknowledge Darrick Lee, Kathryn Hess, Julius von Rohrscheidt, Katharina Limbeck, and Alexandros Keros for providing useful feedback on an early manuscript. They also wish to extend their thanks to the anonymous reviewers and the area chair, who believed in the merits of our work.

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

## A Differential Forms in $\mathbb{R}^n$ Background

In this section, we provide a basic background in the theory of differential forms. The more complete references see Lee (2003) for basic differential geometry and Jost (2017) for Riemannian geometry of $k$-forms. Moreover, see Tao (2009) for a short but highly illuminating article on the intuition behind differential forms.

**Tangent and Cotangent Bundles on $\mathbb{R}^n$.** In $\mathbb{R}^n$, the *tangent space* $T_p(\mathbb{R}^n)$ at a point $p \in \mathbb{R}^n$ is the vector space space spanned by the partial derivatives $\partial/\partial x_i(p)$. The *cotangent space* at $p$ is the linear dual $T^*(\mathbb{R}^n)$ of the tangent space; i.e. the space of linear maps $\mathrm{Hom}_{\mathbb{R}}(T_p(\mathbb{R}^n), \mathbb{R})$. Note that both spaces are isomorphic to $\mathbb{R}^n$. The *tangent bundle* is the space $T(\mathbb{R}^n) = \sqcup_p T_p(\mathbb{R}^n)$ consisting of gluing all tangent space together, where the topology is induced by the projection map $\pi : T(\mathbb{R}^n) \to \mathbb{R}^n$. Likewise, the *cotangent bundle* is the space $T^*(\mathbb{R}^n) = \sqcup_p T_p^*(\mathbb{R}^n)$.

The space of vector fields $\mathfrak{X}(\mathbb{R}^n)$ over $\mathbb{R}^n$ is the space of sections of the tangent bundle; that is, maps $v : \mathbb{R}^n \to T_*(\mathbb{R}^n)$ such that $\pi \circ v = id_{\mathbb{R}^n}$. Vector fields decompose into the form $\sum_i v_i(p)\partial/\partial x_i(p)$, where $v_i : \mathbb{R}^n \to \mathbb{R}$. The space of 1-forms $\Omega^1(\mathbb{R}^n)$ over $\mathbb{R}^n$ consists of the sections $\omega : \mathbb{R}^n \to T^*(\mathbb{R}^n)$ of the cotangent bundle.

**Riemannian Structure on $\mathbb{R}^n$.** The space $\mathbb{R}^n$ is a Riemannian manifold. That is, it has a non-degenerate bilinear form

$$\langle -, - \rangle_p : T_p(\mathbb{R}^n) \otimes T_p(\mathbb{R}^n) \to \mathbb{R}. \tag{12}$$

The inner product is defined on by linear extension of the formula

$$\langle \partial/\partial x_i(p), \partial/\partial x_j(p) \rangle_p = \langle x_i, x_j \rangle \tag{13}$$

where the second inner product is the standard inner product on $\mathbb{R}^n$. Over $\mathbb{R}^n$, the spaces of vector fields and 1-forms are isomorphic via the *sharp isomorphism*

$$\# : \mathfrak{X}(\mathbb{R}^n) \to \Omega^1(\mathbb{R}^n) \tag{14}$$

$$\sum_i v_i \partial/\partial x_i \mapsto \sum_i v_i dx_i \tag{15}$$

where $dx_i$ is the 1-form which is locally $dx_i(p) = \langle -, \partial/\partial x_i(p) \rangle_p : T_p(\mathbb{R}^n) \to \mathbb{R}$.

**Exterior Products.** Let $V$ be a real vector space. Recall that an alternating $k$-covector on $V$ is a map

$$\alpha : \bigotimes_k V \to \mathbb{R} \tag{16}$$

that is alternating with respect to permutations. That is, that

$$\alpha(v_1, v_2, \ldots, v_k) = (-1)^{sgn(\tau)} \alpha(v_{\tau(1)}, v_{\tau(2)}, \ldots, v_{\tau(k)}). \tag{17}$$

The $k$-th exterior product $\Lambda^k(V^*)$ over $V$ is the space of alternating $k$-covectors.

**$k$-Forms.** The $k$-th exterior product of the cotangent bundle $\Lambda^k T^*(\mathbb{R}^n)$ is the tensor bundle defined by locally taking the $k$-th exterior product $\Lambda^k T_p^*(\mathbb{R}^n)$ of each cotangent space. A *differential $k$-form* $\omega \in \Omega^k(\mathbb{R}^n)$ over a $\mathbb{R}^n$ is a smooth section of the $k$-th exterior product of the cotangent bundle $\Lambda^k T^* M$. The space of forms $\Omega^*(\mathbb{R}^n)$ of any dimension has an algebra structure given by the wedge product. The wedge product is multi-linear and satisfies anti-commutativity

$$dx_i \wedge dx_j = -dx_j \wedge dx_i \tag{18}$$

as well as permutation equivariance

$$dx_{i_1} \wedge \ldots \wedge dx_{i_k} = (-1)^{sgn(\tau)} dx_{i_{\tau(1)}} \wedge \ldots \wedge dx_{i_{\tau(k)}} \tag{19}$$

for permutations $\tau$. As described in the body of the paper, $k$-forms have a canonical monomial decomposition given by

$$\omega = \sum_I \alpha_I dx_I. \tag{20}$$

where $dx_I = dx_{i_1} \wedge \ldots \wedge dx_{i_k}$ for a multi-index $I = (i_1, \ldots, i_k)$ and scaling maps $\alpha_I : \mathbb{R}^n \to \mathbb{R}$.

**Types of $k$-forms.** Differential $k$-forms are $k$-forms where the scaling maps $\alpha_I$ are smooth; this is equivalent to the condition $\omega$ that is a smooth section of $k$-th exterior product of the cotangent bundle. When working with both neural $k$-forms and over non-compact spaces like $\mathbb{R}^n$, we often need to define other types of forms.

1. A $k$-form $\omega \in \Omega_c^k(\mathbb{R}^n)$ is *compactly supported* whenever each of the $\alpha_I$ are compactly supported.

2. A $k$-form $\omega \in L_2\Omega^k(\mathbb{R}^n)$ is $L_2$ whenever each $\alpha_I$ is square integrable.

3. A $k$-form $\omega \in \Omega_{PL}^k(\mathbb{R}^n)$ is *piecewise linear* if there exists a triangulation of $\mathbb{R}^n$ such that each $\alpha_I$ is piece-wise smooth over the triangulation.

**Inner Products on $k$-forms.** The choice of Riemannian metric on $\mathbb{R}^n$ extends to an inner product on the $k$-th exterior product of the cotangent space by

$$\langle -, - \rangle_p : \Lambda^k T_p^*(\mathbb{R}^n) \otimes \Lambda^k T_p^*(\mathbb{R}^n) \to \mathbb{R}$$
$$\langle \omega_1 \wedge \ldots \wedge \omega_k, \eta_1 \wedge \ldots \wedge \eta_k \rangle_p \mapsto \det\langle \omega_i, \eta_j \rangle_{i,j}.$$

where $\omega^i, \eta_j \in \Omega^1(\mathbb{R}^n)$. This induces an inner product over the compactly supported $k$-forms $\Omega_c^k(M)$ by integration

$$\langle \omega_1 \wedge \ldots \wedge \omega_k, \eta_1 \wedge \ldots \wedge \eta_k \rangle = \int_{p \in \mathbb{R}^n} \langle \omega_1 \wedge \ldots \wedge \omega_k, \eta_1 \wedge \ldots \wedge \eta_k \rangle_p \, dVol(\mathbb{R}^n) \quad (21)$$

against the volume form $Vol(\mathbb{R}^n)$ on $\mathbb{R}^n$.

**Orthonormal Coframes.** Equip $\mathbb{R}^n$ with the usual Riemannian structure, the compactly supported $k$-forms $\Omega_c^k(\mathbb{R}^n)$ with the inner product $\langle -, - \rangle_\Omega$ described above and induced norm

$$\|\omega\|_\Omega^2 = \langle \omega, \omega \rangle_\Omega$$

for $\omega \in \Omega_c^k(\mathbb{R}^n)$. With this structure, the monomial $k$-forms form an orthornomal coframe in the sense that

$$\langle dx^I, dx^{I'} \rangle_p = \begin{cases} 1 & \text{if } I = I' \\ 0 & \text{else} \end{cases} \quad (22)$$

for each $p \in \mathbb{R}^n$.

**Integration of $k$-forms.** Following (Taylor, 2006), we define integration of monomial $k$-forms $\omega = \sum_I \alpha_I dx_I$ then extend via linearity to arbitrary $k$-forms. Let $\phi : \Delta^k \to \mathbb{R}^n$ be a smooth map and coordinatize $\Delta^k$ with $(t_1, t_2, \ldots, t_k)$ as above.

The *pullback* map of $\phi$ is a map $\phi^* : \Omega(\mathbb{R}^n) \to \Omega(\Delta^k)$ taking forms on $\mathbb{R}^n$ to forms on $\Delta^k$. In coordinates, the pullback $\phi^*\omega \in \Omega^k(\Delta^k)$ of $\omega$ along $\phi$ is defined by the formula

$$\phi^*\omega = \sum_I \alpha_I(\phi^* dx_{i_1}) \wedge (\phi^* dx_{i_2}) \wedge \ldots \wedge (\phi^* dx_{i_k}) \quad (23)$$

where

$$\phi^* dx_i = \sum_j \frac{\partial \phi_i}{\partial t_j} dt_j \quad (24)$$

and $\phi_i$ is the $x_i$-component of $\phi$. Note that by the monomial decomposition in Eq. (3), the pullback can be written as

$$\phi^*\omega = f dt_1 \wedge dt_2 \wedge \ldots \wedge dt_k \quad (25)$$

for some smooth function $f : \Delta^k \to \mathbb{R}$.

We define the integral $\int_\phi$ to be the standard Riemann integral

$$\int_\phi \omega := \int_{\Delta^k} f dt_1 dt_2 \ldots dt_k. \quad (26)$$

The function $f$ can be computed explicitly by unwinding 23 using the algebraic relations 18 and 19. Namely, for the monomial $k$-form $\omega = \alpha_I dx_I$ we have

$$\phi^*\omega = \alpha_I(\phi)\Big( \sum_j \frac{\partial \phi_{i_1}}{\partial t_j} dt_j \Big) \wedge \ldots \wedge \Big( \sum_j \frac{\partial \phi_{i_k}}{\partial t_j} dt_j \Big) \tag{27}$$

$$= \alpha_I(\phi)\Bigg( \sum_\tau sgn(\tau) \frac{\partial \phi_{i_1}}{\partial t_{\tau(1)}} \cdots \frac{\partial \phi_{i_k}}{\partial t_{\tau(k)}} \Bigg) dt_1 \wedge \ldots \wedge dt_k \tag{28}$$

$$= \alpha_I(\phi)\varepsilon^I(D\phi) dt_1 \wedge \ldots \wedge dt_k \tag{29}$$

The above calculation in conjunction with 26 recovers the formula for the integral

$$\int_\phi \omega = \sum_I \int_{\Delta^k} \alpha_I(\phi)\varepsilon^I(D\phi) dt \tag{30}$$

of a general $k$-form $\omega$ given in 5.

**Remark 8.** In the special case that $\phi$ is an affine map, then the Jacobian is expressed as

$$D\phi_{i,j} = \Big[ \phi^i(t_j) - \phi^i(t_0) \Big]_{i,j}$$

**Properties of Integration.** Throughout the paper, we refer to the linearity and orientation equivariance of integration of forms over simplices. These are a consequence of the following theorem.

**Proposition 9** ((Lee, 2003), 16.21). Suppose $M$ is an $n$-manifold with corners and $\omega, \eta \in \Omega^n(M)$ are smooth $n$-forms. Then

1. For $a, b \in \mathbb{R}$ we have

$$\int_M a\omega + b\eta = a \int_M \omega + b \int_M \eta$$

2. Let $-M$ denote $M$ with the opposite orientation. Then

$$\int_{-M} \omega = - \int_M \omega.$$

## B  INTEGRATION OF $k$-FORMS IN PRACTICE

We give further practical details on how to approximate integrals in the case of 2-forms. For more detailed and accessible introductions we recommend the following references (Tao, 2009; Taylor, 2006).

**Explicit computations for $2$-forms.** In Section 5 we use integration of 2-forms on 2-dimensional simplicial complexes in order to classify surfaces. We will derive an explicit method to approximate the integral of 2-forms in $\mathbb{R}^n$ over an embedded 2-simplex based on the theory above.

Recall that a 2-form $\omega \in \Omega^2(\mathbb{R}^n)$ can be written in coordinates as

$$\omega = \sum_{0 \leq i < j \leq n} \alpha_{i,j} dx_i \wedge dx_j, \tag{31}$$

where $\alpha_{i,j} : \mathbb{R}^n \longrightarrow \mathbb{R}$ are smooth maps. We consider a map $\phi : \Delta^2 \longrightarrow \mathbb{R}^n$ giving the embedding of the standard 2 simplex in $\mathbb{R}^n$. Using the expression from Equation 26:

$$\int_{\phi(\Delta^2)} \omega = \int_{\Delta^2} \phi^*(\omega)$$

we will show how to integrate $\omega$ on the embedded simplex $\phi(\Delta^2)$.

Using local coordinates to express the embedding $\phi(t_1, t_2) = (x_1, \ldots, x_n)$ we can write out explicitly the pullback of a 2-form $\omega \in \Omega^2(\mathbb{R}^n)$ via the map $\phi$:

$$
\phi^*\Big( \sum_{0 \le i < j \le n} \alpha_{i,j} dx_i \wedge dx_j \Big) = \sum_{0 \le i < j \le n} (\alpha_{i,j} \circ \phi)\Big(\frac{\partial x_i}{\partial t_1} dt_1 + \frac{\partial x_i}{\partial t_2} dt_2\Big) \wedge \Big(\frac{\partial x_j}{\partial y_1} dt_1 + \frac{\partial x_j}{\partial t_2} dt_2\Big)
$$

$$
= \sum_{0 \le i < j \le n} (\alpha_{i,j} \circ \phi)\Big(\Big(\frac{\partial x_i}{\partial t_1} dt_1 \wedge \frac{\partial x_j}{\partial t_2} dt_2\Big) + \Big(\frac{\partial x_i}{\partial t_2} dt_2 \wedge \frac{\partial x_j}{\partial t_1} dt_1\Big)\Big)
$$

$$
= \sum_{0 \le i < j \le n} (\alpha_{i,j} \circ \phi)\Big(\underbrace{\frac{\partial x_i}{\partial t_1} \frac{\partial x_j}{\partial t_2} - \frac{\partial x_i}{\partial t_2} \frac{\partial x_j}{\partial t_1}}_{A}\Big) dt_1 \wedge dt_2 \tag{32}
$$

In the computational context that we consider we want to integrate a 2-form on the 2-simplices of a complex embedded in $\mathbb{R}^n$. We set some notation here, each 2-simplex $\sigma \in X$ is the image of an affine map $\phi(t) = \Phi t + b : \Delta^2 \longrightarrow \mathbb{R}^n$, where $\Phi = D\phi = [\phi_{i,1}, \phi_{i,2}]_{0 \le i \le n}$ is a $(n \times 2)$-matrix. Since $\phi$ is an affine map the term $A$ in 32 becomes

$$
A = \varepsilon^{(i,j)}(D\phi) = \det \Phi_i^j, \tag{33}
$$

with $\Phi_i^j$ denoting the $(2 \times 2)-$submatrix of $\Phi$ corresponding to the rows $i$ and $j$.

Putting all this together we obtain:

$$
\phi^*(\omega) = \sum_{0 \le i < j \le n} \alpha_{i,j}(\phi) \det \Phi_i^j dt_1 \wedge dt_2. \tag{34}
$$

So the integration becomes

$$
\int_{\phi(\Delta^2)} \omega = \int_{\Delta^2} \phi^*(\omega) = \int_{\Delta^2} \underbrace{\sum_{0 \le i < j \le n} \alpha_{i,j}(\phi) \det \Phi_i^j}_{g} dt_1 \wedge dt_2. \tag{35}
$$

Computationally we cannot perform exact integration of 2-forms so in practise we approximate the integral on $\Delta^2$ above by a finite sum, as is done with Riemann sums in the classical case. In general this is how the `VolApprox` function is defined. To do so the first step is to subdivide the domain of integration $\Delta^2$ into a collection $\mathcal{S}$ of smaller simplices with vertices denoted by $v$. Then the integral is approximated by summing the average value of the map $g$ on each simplex of the subdivision:

$$
\int_{\Delta^2} g \, dt_1 \wedge dt_2 \approx \sum_{s \in \mathcal{S}^2} \frac{1}{3} \sum_{v \in s} g(v) \cdot vol(s). \tag{36}
$$

An explicit method of this sort can be easily generalized to higher dimensional simplices in order to compute integrals of $k$-forms on $k$-dimensional simplicial complexes.

## C    PROOFS

**Universal Approximation I.**    We start with the proof of Proposition 3. First, recall the well-known following universal approximation theorem for neural networks. We cite here a version appearing in (Pinkus, 1999).

**Theorem 10** (Universal Approximation Theorem, Thm 3.1 (Pinkus, 1999)). *Let $\sigma \in C(\mathbb{R}, \mathbb{R})$ be a non-polynomial activation function. For every $n, \ell \in \mathbb{N}$, compact subset $K \subset \mathbb{R}^n$, function $f \in C(K, \mathbb{R}^\ell)$ and $\epsilon > 0$ there exists:*

- *an integer $j \in \mathbb{N}$;*
- *a matrix $W_1 \in \mathbb{R}^{j \times n}$;*

- *a bias vector $b \in \mathbb{R}^j$ and*

- *a matrix $W_2 \in \mathbb{R}^{\ell \times j}$*

*such that*

$$\|f - g\|_\infty < \varepsilon$$

*where $g$ is the single hidden layer MLP*

$$g(x) = W_2 \sigma(W_1 x + b).$$

**Theorem 3.** *Let $\alpha \in C(\mathbb{R}, \mathbb{R})$ be a non-polynomial activation function. For every $n \in \mathbb{N}$ and compactly supported $k$-form $\eta \in \Omega_c^k(\mathbb{R}^n)$ and $\epsilon > 0$ there exists a neural $k$-form $\omega^\psi$ with one hidden layer such that*

$$\|\omega^\psi - \eta\|_\Omega < \epsilon.$$

*Proof.* Denote the scaling functions of $\eta$ by

$$\eta = \sum \eta_I dx_I.$$

Since $\eta$ is compactly supported, each $\eta_I$ is compactly supported over some domain $D$. By 10, for any $\varepsilon > 0$ there exists a one layer MLP $\psi : \mathbb{R}^n \to \mathbb{R}^{\binom{n}{k}}$ such that

$$\|\psi - \bigoplus_I \eta_I\|_\infty < \varepsilon/Vol(D)^{1/2}.$$

Using the orthogonality form *Eq. (22)*, we have that

$$\|\omega^\psi - \eta\|_p^2 = \langle \omega^\psi - \eta, \omega^\psi - \eta \rangle_p = \sum_I (\psi_I(p) - \eta_I(p))^2 = \|\psi(p) - \oplus_I \eta_I(p)\|_{\mathbb{R}^{\binom{n}{k}}}^2 \leq \varepsilon^2/Vol(D).$$

Integrating the above, we attain

$$\|\omega^\psi - \eta\|_\Omega^2 \leq \int_D \langle \omega^\psi - \eta, \omega^\psi - \eta \rangle_p dVol(\mathbb{R}^n)$$

$$< \varepsilon^2/Vol(D) \int_D dVol(\mathbb{R}^n)$$

$$= \varepsilon^2$$

proving the result. $\qquad \square$

**Equivariance.** Integration is linear in both $k$-forms (Lee (2003), 16.21) and embedded simplicial chains 7 in the sense that it defines a bilinear pairing

$$\int : C_k(\mathcal{S}; \mathbb{R}) \otimes \Omega^k(\mathbb{R}^n) \to \mathbb{R} \tag{37}$$

for some embedded simplicial complex $\phi : \mathcal{S} \to \mathbb{R}^n$. This property directly implies the kind of multi-linearity described in Proposition 6. We present a proof here for completeness.

**Proposition 6** (Multi-linearity). *Let $\phi : \mathcal{S} \to \mathbb{R}^n$ be an embedded simplicial complex. For any matrices $L \in M^{m' \times m}(\mathbb{R})$ and $R \in M^{\ell \times \ell'}(\mathbb{R})$ we have*

$$X_\phi(L\beta, \omega R) = LX_\phi(\beta, \omega)R \tag{38}$$

*Proof.* On the left we have

$$X_\phi(L\beta, \omega)_{i,j} = X\Big(\sum_k L_{1,k}\beta_k, \ldots, \sum_k L_{m',k}\beta_k, \omega\Big)_{i,j} \tag{39}$$

$$= \int_{\sum_k L_{i,k}\beta_k} \omega_j \tag{40}$$

$$= \sum_k L_{i,k} \int_{\beta_k} \omega_j \tag{41}$$

$$= \Big[LX_\phi(\beta, \omega)\Big]_{i,j}. \tag{42}$$

Similarly, on the right we have

$$X_\phi(\beta, \omega R)_{i,j} = X_\phi\Big(\beta, \sum_k R_{k,1}\omega_k, \ldots, \sum_k R_{k,\ell'}\omega_k\Big)_{i,j} \tag{43}$$

$$= \int_{\beta_i} \sum_k R_{k,j}\omega_k \tag{44}$$

$$= \sum_k R_{k,j} \int_{\beta_i} \omega_k \tag{45}$$

$$= \Big[X_\phi(\beta, \omega)R\Big]_{i,j} \tag{46}$$

$\square$

## D  ADDITIONAL EXPERIMENTS

### D.1  CONVOLUTIONAL 1-FORM NETWORKS

**Convolution and Equivariance.**    Beyond orientation and permuation equivariance, the node embeddings themselves may be equivariant with respect to a group action on the feature space. The class of *geometric graph neural networks* Han et al. (2022) were designed specifically to deal with such equivariances. The lesson from image processing is that translation equivariance can be mitigated via *convolution*. In this paradigm processing embedded (oriented) graphs with learnable, 'template' 1-forms is directly analogous to processing images with learnable convolutional filters. To make this connection precise, we will perform a small example to classify oriented graphs.

**Synthetic Data.**    In Figure 5, there are two classes of cycle and star graphs. The cycle graphs have clockwise orientation and the star graph are oriented so edges point inwards. Each data-point is a simplicial complex representing the disjoint union of either three cycle or star graphs which are randomly recentered around the unit square. The chains on each complex are initialised using the standard oriented 1-simplices as a basis.

**Architecture.**    We initialise a neural network $\psi : \mathbb{R}^2 \to \mathbb{R}^{2\times 2}$ with a single 64-dimensional hidden layer and `ReLU` activation which corresponds to two feature 1-forms. To perform a convolutional pass, we first discretize the unit square to produce a set of translations. At each translation, we restrict the embedded graph to the subgraph within a neighbourhood and weight it by the local node density approximated by a standard kernel density estimator. This is equivalent to translating the 1-form by the corresponding (negative) vector in $\mathbb{R}^2$ and integrating over a small neighbourhood (whose area is a hyperparameter). Integration produces an integration matrix at each point with two columns, and taking the column sum represents the oriented integral of the two convolutional filters against the local neighbourhood of the oriented graph. `CrossEntropyLoss` is calculated by summing the integrals over all translations and applying `softmax`.

**Interpreting the Results.**    The integrals are shown in Figure 5 as a colouring of each point in the grid of translations. We plot the learned 'template' 1-forms next to an example of their respective classes. By the construction of the objective function, the algorithm is trying to maximise the sum of the integrals across all translations within the class. As in the synthetic paths example, the learned filters successfully capture the interpretable, locally relevant structure; the edges in each class resemble flow-lines of the learned vector field in different local neighbourhoods around the relevant translations.

### D.2  VISUALISING SIMPLICIAL LAPLACIAN 1-EIGENVECTORS

In this example, we show how we can use neural 1-form to visualise the 1-eigenvectors of the simplicial Laplacian[8] of a Rips complex in $\mathbb{R}^2$ (Figure 6). The idea is that we start with a collection of eigenvectors of simplicial 1-cochains, and learn 1-forms which integrate to them. We see the results

---

[8]See Appendix for definitions.

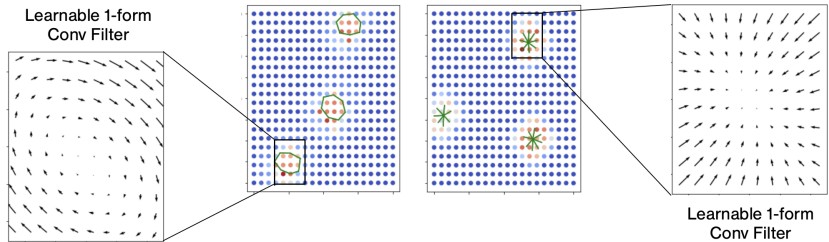

Figure 5: A convolutional 1-form network with the learned convolutional filters forms.

in Figure 6b, where the three eigenvectors belong to the harmonic, gradient-like and curl components of the simplicial Hodge decomposition respectively.

First we generate a point cloud in $\mathbb{R}^2$ as a noisey approximation of a circle, then take the Rips complex for a fixed parameter $\varepsilon$. We then calculate the first 9 eigenvectors of the 1-simplicial Laplacian, sorted by increasing eigenvalue, and store them as columns of a matrix $Y$ with respect to the standard basis $\beta$. We initialise 9 neural 1-forms

$$\omega = (\omega_1, \omega_2, \ldots, \omega_9) \in \bigoplus_\ell \Omega^1_{PL}(\mathbb{R}^2)$$

on $\mathbb{R}^2$ using a ReLU activation function. Our loss function is then

$$\mathcal{L}(\omega) = \|X_\phi(\beta, \omega) - Y\|$$

where the norm is the standard matrix norm, and $X_\phi(\beta, \omega)$ is the evaluation matrix yielded from integrating $\omega$ over $\beta$. When $\mathcal{L}$ is small, the 1-forms over $\mathbb{R}^2$ will correspond to the selected simplicial eigenvectors when integrated over the complex.

Figure 6 shows the results—one notes that the different 1-forms appear to coalesce around geometric features in the underlying point cloud. Additionally, it is important to observe that the Laplacian 1-eigenvectors can be categorized into three distinct classes depending on their membership within the components of the Hodge decomposition:

1. The first class consists of eigenvectors that belong to the image of the adjoint of the differential operator $d^1$,

2. The second class comprises eigenvectors that reside within the kernel of the Laplacian operator $\Delta^1$,

3. Lastly, the third class includes eigenvectors that are part of the image of the differential operator $d^0$.

These classes correspond, in the context of simplicial structures, to analogues of rotational, harmonic, and gradient-like vector fields found in Riemannian manifolds. By referring to Figure 6b, one can visually identify the class to which each eigenvector belongs based on whether or not there exists rotational structure. We have color-coded these as green, red, and blue, respectively.

## E   EXPERIMENTAL SETUP AND IMPLEMENTATION DETAILS

### E.1   SYNTHETIC PATH CLASSIFICATION

**Synthetic Path Classification.**   We initialise neural 1-forms $\omega_1, \omega_2, \omega_3 \in \Omega^1_{PL}(\mathbb{R}^2)$ with ReLU activation for each of the three classes. Each path $p$ is represented as an oriented simplicial complex, where the orientation is induced by the direction of the path. Letting $\beta$ be the standard basis, the three 1-forms generate an evaluation matrix $X_\phi(\beta, \omega)$ whose entries are the integration of the 1-form $\int_{e_i} \omega_j$ against each 1-simplex in the paths.

The readout function sums the entries in each column, which by the linearity of integration, represents the path integral along $p$ against each $\omega_i$. In short, each path $p$ is represented as a vector in $\mathbb{R}^3$ using

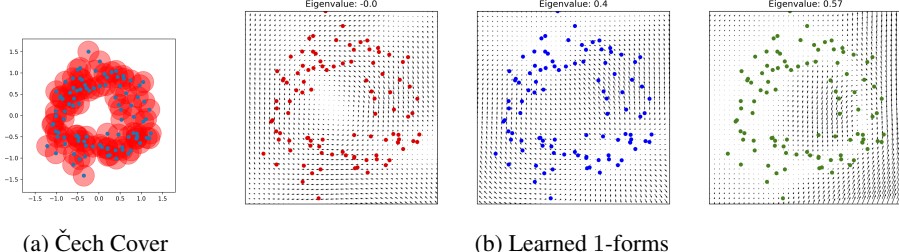

(a) Čech Cover                    (b) Learned 1-forms

Figure 6: Approximating the 1-eigenvectors of the simplicial Laplacian with learnable 1-forms using the Čech Cover of a point cloud.

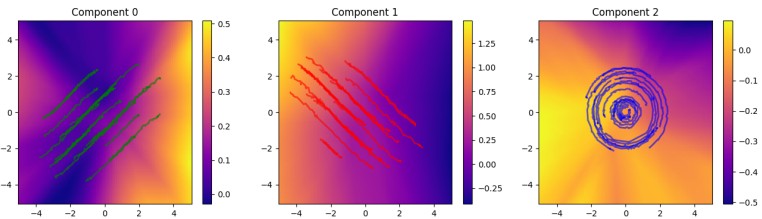

Figure 7: Learned 0-forms for Synthetic Path Classifcation.

the map

$$p \mapsto (\int_p \omega_1, \int_p \omega_2, \int_p \omega_3) \in \mathbb{R}^3. \tag{47}$$

We then use cross-entropy loss between this vector and the class vector. In this design, each 1-form corresponds to a class, and a path should ideally have a high path integral against this form if and only if it belongs to the class.

*Comparison with Neural Networks.* In our synthetic experiment, we test whether edge data is necessary by examining whether we can attain comparable results using only the embedded vertices of the path.

In our context, the right framework to analyse the vertices is a set of 0-forms on $\mathbb{R}^2$ - in other words, scalar functions over $\mathbb{R}^2$. We use the same setup as before, where we initialise three functions $f_1, f_2, f_3 \in \Omega^0_{PL}(\mathbb{R}^2)$ corresponding to the three classes. Integration of the vertices in the path against each of $f_i$ simply corresponds to evaluating $f_i$ at the vertex. Summing the columns of the evaluation matrix then sums up the value of $f_i$ at all points in a path. In this sense, each $f_i$ functions much like an approximated 'density' for the vertices of each class - albeit with negative values.

In Figure 7, we show example paths from each class against the learned scalar function representing that class. In this example, the vertices of paths in each class have a similar density. One sees that the algorithm learns something reasonable, picking out minor fluctuations in density, but struggles overall to separate out the classes. With the same number of parameters and training time, the algorithm is only able to achieve a training accuracy of 37%.

### E.2    SYNTHETIC SURFACE CLASSIFICATION

We consider two classes of synthetic surfaces obtained by embedding a triangulated square in $\mathbb{R}^3$. The embedding of the vertices of the triangulated square are given by functions of the type $\phi_1(x, y) = sin(x) + \epsilon(x, y)$ for the first class and $\phi_2(x, y) = sin(y) + \epsilon(x, y)$ for the second class, where $\epsilon$ is random noise. We initialise two neural 2-forms $\omega_1, \omega_2 \in \Omega^2_{PL}(\mathbb{R}^3)$ with ReLU activation for each of the classes. Letting $\beta$ be the standard basis, the two 2-forms generate an evaluation matrix $X_\phi(\beta, \omega)$ whose entries are the integration of the 2-form $\int_{e_i} \omega_j$ against each 2-simplex in the surfaces. The readout function sums the entries in each column, which represents the integral of each

Class 1                           Class 2

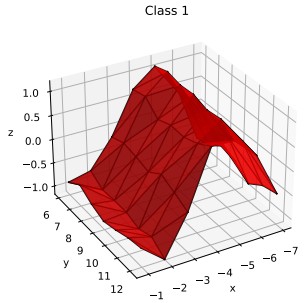 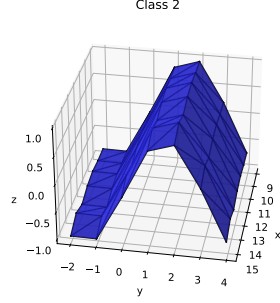

(a) Integral of $\omega_1$ on a representative of the first class.  (b) Integral of $\omega_1$ on a representative of the first class.

Figure 8: Representative surfaces of each classes, colored by the integral of the learned 2-form $\omega_1$ over the surface. The integral of $\omega_1$ over elements of the first class yields positive values whereas the integral of the same form over elements of the second class yields negative values. For the form $\omega_2$ the opposite is true.

$\omega_i$ over the entire surface, thus yielding a representation of the surfaces in $\mathbb{R}^2$ in the following way

$$s \mapsto (\int_s \omega_1, \int_s \omega_2,) \in \mathbb{R}^2. \tag{48}$$

Finally, we use the cross entropy loss function to classify the surfaces into the two classes. In this design, each 2-form corresponds to a class, and a surface should ideally have a high integral against this form if and only if it belongs to the class.

Figure 8 shows the integral of $\omega_1$ on surfaces taken from each of the two classes. This integral is positive for elements of the first class and negative for elements of the second class. Similarly the values for the integral of $\omega_2$ is negative on surfaces of the first class and positive on elements of the second class.

### E.3 GRAPH BENCHMARK DATASETS

For the small graph benchmark datasets (AIDS, BZR, COX2, DHFR, Letter-low, Letter-med, Letter-high), we use a learning rate of $1e - 3$, a batch size of 16, a hidden dimension of 16, and $h = 5$ discretisation steps for all $k$-forms. For our comparison partners, i.e. for the graph neural networks, we use $h$ hidden layers to permit message passing, followed by additive pooling. As a result, all models have roughly equivalent parameter budgets, with our model having access to the *smallest* number of parameters.

**Architectures.** Our model architectures for our comparison partners follow the implementations described in the respective papers (Kipf & Welling, 2017; Veličković et al., 2018; Xu et al., 2019). We make use of the `pytorch-geometric` package and use the classes `GAT`, `GCN`, and `GIN`, respectively. Our own model consists of a *learnable vector field* and a *classifier* network. Letting $H$ refer to the hidden dimension and $D_{\text{in}}$, $D_{\text{out}}$ to the input/output dimension, respectively, we realise the vector field as an MLP of the form `Linear[`$D_{\text{in}}, H$`] – ReLU – Linear[`$H, H/2$`] – ReLU – Linear[`$H/2, D_{\text{out}}$`]`. The *classifier network* consists of another MLP, making use the number of steps $h$ for the discretisation of our cochains. It has an architecture of `Linear[`$h, H$`] – ReLU – Linear[`$H, H/2$`] – ReLU – Linear[`$H/2, c$`]`, where $c$ refers to the number of classes.

**Training.** We train all models in the same framework, allocating at most 100 epochs for the training. We also add *early stopping* based on the validation loss with a patience of 40 epochs. Moreover, we use a learning rate scheduler to reduce the learning rate upon a plateau.

