# OpenReview forum: "Simplicial Representation Learning with Neural $k$-Forms"
_ICLR.cc/2024/Conference — ICLR 2024 poster_

### Official Review · Reviewer_rtvN · 2023-10-22

**Soundness:** 4 excellent
**Presentation:** 2 fair
**Contribution:** 3 good
**Rating:** 8
**Confidence:** 2

**Summary:**

This paper proposes using differential $k$-forms in $\mathbb{R}^n$ to create representations of geometric simplices. Specifically, *neural* $k$-forms are parameterized by MLPs that take geometric information. By then integrating the $k$-forms using the simplicial complex, the authors obtain the *integration matrix*, which can be read out for further processing. The method allows processing simplicial data without the need for (simplicial) message passing. Solid theoretical results are supported with several motivating experiments, opening new directions for future research.

**Strengths:**

* The presented ideas are original, relevant and opening up new directions of research.
* The theory, though not trivially graspable for the typical machine learner, is presented clearly.
* The theory is motivated with convincing experiments.

**Weaknesses:**

* While the content of the paper is naturally foreign for most machine learning researchers, the authors could put a bit more effort into making the content slightly more accessible. I would encourage them to include as much examples as possible, especially since the full 9 pages were not completely used yet. The authors can gain even more space by e.g. leaving out the introduction to MLPs, which should be familiar for the majority of readers.
* Experimental details are highly suppressed, it is not quite clear what the inputs, outputs, and learning objective of the experiments are.

Minor:
* Above eq. 1, you introduce the notation of a convex hull using $v_0, v_1, \dots, v_k \in \mathbb{R}^n$, but before and after you use $v$ and $v_i$ for vertices in $V$. This caused some confusion for me while reading this part. Similarly on page 3 when introduction $k$-vectors.
* Experiments are mostly limited to the synthetic setting.

**Questions:**

* Could you discuss the unique linear map $\phi|_\sigma: \Delta^k \to \mathbb{R}^n$? It is not quite clear how this function for $k < n$ is well-defined?
* At equation 9, I was a bit lost how we define $\phi$. Before eq. 4 we define $\phi: S \to \mathbb{R}^n$, but here $\phi$ seems to take $t \in \Delta^k$ which seems more in line with $\phi|_\sigma$. Did we suppress some arguments here, or are we assuming something about the elements of $S$?
* What are the inputs and outputs to your neural networks?  Are the positions of the node locations input as $(\mathbb{R}^d)^n$ when you have $n$ points, or do you concatenate them?
* Could you elaborate a bit more on how you go from $\mathbb{R}^{\binom{n}{k} \times \ell}$, together with the simplicial complex structure, to the integration matrix $X_\phi(\beta, \omega)$? Specifically, in eq. 13, the sum is over all multi-indices, correct? Do we need to know $\lambda_a$ somehow?
* Could you discuss the advantages of your method over (simplicial) message passing? Why does it yield better empirical results? Should this method replace message passing altogether, or are there limitations?



**Final Remarks**. Admittedly I'm lacking some background knowledge to objectively assess the paper. Despite that, I'm really excited about this project. However, I think if the authors put a bit more effort into conceptualizing some core ideas, the paper would improve considerably.

---

> ### Author Response · Authors · 2023-11-22
> **Official answer to Reviewer rtvN**
>
> We thank the reviewer for their constructive comments. Judging from their comments, it is clear that the reviewer made a concerted effort to understand the mathematical concepts and provided some excellent questions that helped us clarify the presentation of the math.
>
> &nbsp;
>
>
> ### Answer primary weaknesses
>
> &nbsp;
>
> 1. We agree with the reviewers comments that the paper could be made more accessible.  We appreciate the reviewer’s suggestion of cutting out some of the basic ML definitions, since these should be background for the audience.  We have decided in the revision to globally put more emphasis into explaining the ML aspects of the work in the body of the paper. The trade-off is that we were unable to give a more detailed explanation of the mathematical concepts which we leave for the appendix.
>
>     We have done this in two major revisions. Firstly, we have added a new subsection *Architectures and Parameters* which explains in detail how one can generally construct integration matrices from MLPs, and what the parameters and hyper-parameters of the algorithm are. Judging from the majority of the reviews, the pipeline through which the k-forms are actually used in an ML pipeline was a core concept that was poorly conveyed. We hope that this additional section goes some way to remedy this problem.
>
> &nbsp;
>
> 2. Secondly, as suggested by the reviewer, we have greatly expanded the descriptions of the experiments section in the body to include more details of the implementation. Further, we have tried to provide a more guided curation of how to interpret the results. Our hope is that for readers who are put-off by the technical definitions in the body will now have more chance of grasping the natural and meaning of the synthetic experiments and core ideas.
>
> &nbsp;
>
> ### Answer Minor Weaknesses
>
> &nbsp;
>
> 1. Thank you for pointing this out - there were a number of unfortunate typos in the text which we have now clarified.
>
> &nbsp;
>
> 2. We think that the graph-learning tasks - nine in total - constitute a large enough sample to argue that our method can be used in practice. For us, the main goal of the paper is to introduce the core concepts, which are quite different to the usual message-passing architectures used in these kinds of tasks and require the majority of the space in the paper. We agree that the method should be tested on a wider and more rigorous framework. However, we propose that this is beyond the scope of this initial, introductory paper.

---

> > ### Author Response · Authors · 2023-11-22
> > **Official answer to Reviewer rtvN - part 2**
> >
> > ## Answer Questions
> >
> > &nbsp;
> >
> > 1. The embedding of the simplicial complex determines uniquely the position of the nodes, the edges and the k-simplices are uniquely determined by linearly interpolating between the nodes. For any $k<n$ this is uniquely defined since there can be only one line segment between two given points, only one triangle through 3 given points in $\mathbb{R}^n$ no matter how big $n$ is.   However this map would no longer be unique if $k>n$ since there would be no way of embedding a $k$-dimensional object in a lower dimensional space, fortunately this does not happen in practice as the graph features determining the embeddings are high dimensional
> >
> > &nbsp;
> >
> > 2. Indeed this notation was not clear, it has been updated in the paper (see equation 15).
> >
> > &nbsp;
> >
> > 3. The data are the set of embedded oriented simplicial complexes, with chains assigned to them. Since $\lambda_a$ is a coefficient of a chain, it is part of the data. When no chains are provided, they are taken to the standard basis chains corresponding to each simplex. Here, each simplex $\sigma_a$ is mapped to the chain whose coefficient $\lambda_a$ of $\sigma_a$ is 1 with zero coefficients elsewhere.
> >
> >     The learnable part is the MLP. The input of the MLP are points in the ambient feature space $\mathbb{R}^n$, and the output dimensions each correspond to a scaling function (of which there are n choose k) of each for L different differential forms. This is a slightly subtle but important point; the input to the MLP is not the simplicial complex itself, or its specified set of k-chains, but the ambient feature space in which the simplicial complex is embedded. This makes the algorithmic pipeline feel somewhat unfamiliar to most MLP-based approaches.
> >
> >     So, rather than using the simplicial complex as an input to the MLP, we integrate neural $k$-form (MLP) against the simplicial complex to produce an integration matrix representation of the complex and chains. To be explicit, the integral in Equation 15 (revised version) is approximated by selecting a finite set of points t in the simplex and replacing the integral on the RHS with a sum weighted according to the areas in a corresponding subdivision of $\Delta^k$. If the chain is linear combination of simplices $\sum_i \lambda_i \sigma_i$, we just take linear combinations of the above using the coefficients $\sigma_i$. The integral involves calculating the Jacobian of the embedding function at a finite number of places, which is particularly easy for linearly embedded complexes.
> >
> >     The scaling functions alpha correspond to the output dimensions of the MLP. We hope it is clear that this approximated integral is smoothly dependent on the parameters of the MLP. By doing this for each pair of simplex and form we produce the integration matrix, whose entries thus also depend on the parameters in the MLP. Any downstream processing of the integration matrix that leads an objective function will make backpropagation with respect the MLP parameters possible, and that is indeed how our learning process takes place.
> >
> >     We have added this explanation to the body of the text which we hope addresses the reviewer’s questions and clarifies the presentation.
> >
> > &nbsp;
> >
> > 4. We think this is addressed by the previous detailed explanation. Again, the integration matrix is produced by a pairing of the learnable k-form and a data-point (which consists of an embedded simplicial complex with a specified subset of k-chains). And the sum in equation is over the simplices in the simplicial complex against which we are integrating.

---

> > > ### Author Response · Authors · 2023-11-22
> > > **Official answer to Reviewer rtvN - part 3**
> > >
> > > 5. We argue that this method should replace simplicial - and potentially graph - message passing completely, but admit this claim still needs to be fleshed out by many more experiments. Indeed, since the benchmark sets are not well-established it is not clear how to decisively show that one is better than the other except for comparison on existing graph benchmarks. We note that in benchmark tasks such as Ocean Flow in [1] many simplicial NNs get near 100% accuracy, rendering them effectively useless for benchmarking.
> > >
> > >     There are other reasons why we think our method is preferable. Simplicial message passing takes as input a matrix of cochains for each simplicial complex. What is missing is that often these cochains do not correspond to the same global feature defined over the data; so it is difficult to compare them across complexes. We hope in our experiments we have at least shown cases where one can directly interpret the feature 1-forms in a way that seemingly has no analogue in message passing.
> > >
> > >     Finally and most significantly, we believe that this method could potentially address the pervasive over-smoothing problem that occurs in most message passing algorithms. Adding in more and more layers to an MLP will produce a more and more intricate k-form. It seems feasible that, since really this is just an MLP under the hood, these types of networks may enjoy the same depth-scaling effects when applied over larger datasets, rather than oversmoothing. Indeed, we also see the method become more powerful relative the standard message-passing GNNs as we move to the larger datasets in MoleculeNet as opposed to TUDatasets.
> > >
> > > &nbsp;
> > >
> > >
> > > [1]  Bodnar, C., Frasca, F., Wang, Y., Otter, N., Montufar, G.F., Lió, P. & Bronstein, M.. (2021). Weisfeiler and Lehman Go Topological: Message Passing Simplicial Networks. Proceedings of the 38th International Conference on Machine Learning, in Proceedings of Machine Learning Research 139:1026-1037 Available from https://proceedings.mlr.press/v139/bodnar21a.html.

---

> > > > ### Comment · Reviewer_rtvN · 2023-11-23
> > > >
> > > > Thanks for your feedback. I do not have time anymore to consider the revised version. However, I believe the authors made considerable efforts to improve the paper based on their replies. As such, I've promoted my score while keeping a low confidence.

---

### Official Review · Reviewer_gTeC · 2023-10-31

**Soundness:** 3 good
**Presentation:** 2 fair
**Contribution:** 3 good
**Rating:** 6
**Confidence:** 4

**Summary:**

This paper proposed a novel method for representation learning of geometric simplicial complexes, in which a differential form parameterized by a multilayer perceptron is integrated over the simplices of an embedded complex, and these integrals are then read out into a representation of the complex that is not dependent on the size or dimension of the complex. This method is considered in a variety of tasks for processing simplicial complexes and graphs with geometric information.

**Strengths:**

1. The approach proposed by the authors is novel, yet simple, and rooted in a geometric view of learning for simplicial complex data.
2. The proposed method is able to apply one learned function (k-form) to tasks over many geometric simplicial complexes, rather than only learning a function for a given complex.

**Weaknesses:**

1. This paper suffers from vagueness in some parts, particularly in the experiments section. This is my reason for rating the presentation as "fair." Otherwise, the writing of the paper is good.
2. Compared to message-passing schemes, the learned k-forms appear to be highly dependent on the particular geometric embedding of the complex, instead of the intrinsic topology/geometry.
3. The type of tasks considered by the authors are not common ones in the literature, so I am not sure if there is a regime in which the proposed methods can be compared to existing simplicial neural networks, which are largely based on combinatorial, rather than geometric, information. However, there are message-passing graph neural networks that are designed to handle geometric information as well, which the authors did not compare to.

**Questions:**

1. My primary suggestions relate to the writing of the experiments section. There is not sufficient detail in Section 5 to understand each of the experiments. I think it would be better to shorten some of the background material in order to have space for better descriptions of your experiments. In particular:

a. Path classification: are the "paths" that you classify themselves simplicial complexes? I.e., is the idea to represent each path as a 1-d simplicial complex embedded in $\mathbb{R}^2$, and then integrate against the simplices of the path? I see this to be the case in the appendix, but these details should really be in the main body of the paper for the purpose of readability.

b. Visualizing simplicial Laplacian 1-eigenvectors: the description of this is far too short to understand what is going on here, and the utility of this is questionable. Comparing to the eigenvectors of the simplicial Laplacian is reasonable for the purpose of identifying (co)homological features of the complex, but wouldn't it be better to use neural k-forms to get a proxy for the intrinsic eigenfunctions of some manifold embedded in $R^n$ from which a complex is formed? This experiment needs a lot more motivation.

c. Synthetic surface classification: again, having a description of the dataset in the main body of the paper is needed for readability here. It is fine to leave some details in the appendix, but the body of the paper should be sufficient to get a good understanding of what is going on.

2. I have some questions about invariants that can be incorporated into these networks. As I commented in the weaknesses section, the function learned by neural k-forms is highly dependent on particular geometric embeddings of the simplicial complexes in Euclidean space, as opposed to the intrinsic geometry of the complex. For instance, if I apply a translation to the embedding of the complex, I am likely to get a completely different result when integrating the neural k-form against it. Do you have any comments or insights on handling issues like this? This relates also to your experiment on classifying molecular graphs: the graphs can be geometrically embedded in $\mathbb{R}^3$, but is there a canonical rotation that these embeddings should have? Two identical molecules might be embedded in different ways that yield different results by the network, which is not a property that a message-passing network suffers from. Perhaps a group symmetry should be incorporated for this issue. See the following reference, for instance:

Han, J., Rong, Y., Xu, T., & Huang, W. (2022). Geometrically equivariant graph neural networks: A survey. arXiv preprint arXiv:2202.07230.

I think comparison between neural k-forms and geometric GNNs should be made, in order to yield a fairer picture of where your method stands for graph classification tasks.

3. Can you please comment on whether or not simplicial complexes with non-geometric features could be incorporated into the method you propose? For instance, if the geometric complex not only has an embedding, but also a particular (co)chain supported on it supplied, how would one incorporate that information into a learning pipeline?

---

> ### Author Response · Authors · 2023-11-22
> **Official Anser to Reviewer gTeC - part 1**
>
> We thank the reviewer for their extremely in-depth and constructive review. In particular, the reviewer’s feedback about the equivariances and embeddings prompted to better clarify that our model explicitly ignores node/simplice embedding information which is equivariant with respect to actions on the ambient space. We realise we did a poor job of conveying this in the original version and have since amended this in the revision.
>
> &nbsp;
>
> ### Answer to Weaknesses
> &nbsp;
>
> 1. We note that this was indeed a common criticism amongst the reviewers. We have taken the advice and decided to greatly expand the details of the experiments section.
>
> &nbsp;
>
> 2. Our basic premise is that a simplicial complex is itself only a topological object and has no intrinsic geometry. Indeed, for a combinatorial simplicial complex, the standard ‘geometric realisation’ coming from algebraic topology without an embedding is only a topological space. We argue that in simplicial and graph neural networks the geometry of the simplicial complex is determined by how it is embedded in $\mathbb{R}^n$: a space that has a Riemannian structure and notions of distance, angle and geometry.
>
>     Hence our model is dependent on both the particular geometric embedding and the intrinsic combinatorial structure of the graphs and complexes. Indeed the integration matrices, and therefore the cochains and the output as well, are determined by the connections (edges, k-simplices) between the nodes since these connections tell us where exactly to integrate the k-forms. The same set of node positions but with very different connections will yield different cochains
>
>     Most graph message-passing schemes have some non-equivariant component of the node features - for example the atomic weight, one hot encoded atom type, valence in the molecular datasets. In this work, we are interested in only the geometry of these types of features. For example, even when we ignore the ‘positions’ of the atoms in the molecular graph, we claim that there is still interesting geometry in the node feature space. In the message-passing paradigm, the topology and combinatorics of the complex are then incorporated into how messages are passed. For us, the difference is that the topology and combinatorics are represented as determining the regions of integration (k-simplices) in the embedding space over which learnable k-forms are integrated.
>
>     The question of how to adapt our model to deal with node/complex embeddings that are equivariant in the ambient space - such as the ‘position’ of atoms in a molecule - is related. For this we ask the reviewer to see our detailed response to Question 2 and the added example on ‘convolutional 1-form networks’ in the revised appendix *Additional Experiments*.
>
> &nbsp;
>
> 3. At least for us the TUDataset for experiments in Table 1 and MoleculeNet datasets in Table seem relatively common in the literature and are recommended in the (Dwivedi et al., 2023) reference. We would be open to suggestions of further datasets on which to evaluate our model given we have an implementation of the architecture ready.
>
>     We have decided to follow the reviewer’s advice on the point of adding an extra point of comparison with a Geometric GNN; namely, the equivariant GNN of (Santorras et. al 2021). As we had poorly conveyed and now stressed in the revised version, our method the equivariant component of node coordinates (such as the positional information) is always omitted from our model.
>
>     The TU datasets in Table 1 have both positional and non-equivariant features from the 1-hot encoding of the atom type. For the GCT, GAT, GIN and our model, the position features were suppressed; the EGNN was provided with the additional equivariant information. We found that the equivariant model did not perform significantly better than the models without the positional information. Curiously the out-of-the-box baseline EGNN model also had significantly more parameters than the others.
>
>     For the larger MoleculeNet dataset, we selected only datasets without equivariant, positional node embeddings so that we could apply our model. While we ran out of time to do this before the revision, we would be happy to run further experiments on larger datasets in MoleculeNet against EGNN that have positional coordinates in addition to non-equivariant coordinates if the reviewer thinks it would significantly improve the paper. The main thrust of this experiment was to show that our method scales drastically better than the standard (non-geometric) GAT, GIN and GCN models.

---

> > ### Author Response · Authors · 2023-11-22
> > **Official Anser to Reviewer gTeC - part 2**
> >
> > ### Answer Questions
> > &nbsp;
> >
> > 1. We have added a number of details to all of the experiments in the experiments section. Further, we have added an ‘Architectures and parameters’ section which we think clarifies the general framework underlying all of the experiments.
> > &nbsp;
> >
> >     a. A piecewise linear path in $\mathbb{R}^n$ can be thought of as a simplicial complex with an embedding. The 1-simplices are linear maps from the interval starting at the i-th embedded node and ending at the (i+1)-st embedded node. The corresponding 1-chain consisting of the sum of all oriented 1-simplices represents the path as a chain, and the integral of this chain against a 1-form corresponds to the path integral. This has been added to the paper to clarify for the reader.
> >
> >     b. On reflection, we agree with the reviewer’s critique of this experiment and have decided to move it to the appendix to make space for additional experimental details in the others.
> >
> >     c. We agree the presentation of this example was too minimalist, the example about surface classification has been expanded to contain more details in the main body of the paper.
> > &nbsp;
> >
> > 2. The reviewer makes an excellent point about equivarance/invariant of ‘positional’ embeddings. In the examples of molecular graphs, ‘positional embeddings’ that differ by an isometry on $\mathbb{R}^n$ should be identified - in other words vector representations of the embedded simplicial complex should be isometry invariant over these embedding dimensions. To address this, we have brought forward some of our future work to give an example of a ‘Convolutional 1-form network’ as a newly added example of the Additional Experiments section in the appendix.
> >
> >     Much like image analysis, we choose a family of translations and record the integral of the sub-simplicial complex in a local neighbourhood of each translation. The output is a real-valued function over a finite subset T’ of the translation group T on $\mathbb{R}^n$. Both the space of embedded simplicial complexes in $\mathbb{R}^n$ and the space of real-valued functions over T have a natural action of the translation group; the assignment of an embedded simplicial complex to the output convolution function over T is equivariant with respect to these actions. Integration of the function over the translation elements exists whenever both forms and simplicial complexes are compactly supported and represent a translation invariant quantity. The finite sum over the subset of translations we perform in the experiment is an approximation of this invariant quantity. The above could feasibly fit into a larger scheme of convolutions where we have the action of a Lie group G on $\mathbb{R}^n$, and we plan to explore this with other isometry sub-groups in future work where will incorporate it into benchmarks. Due to space constraints, we believed that a full exposition and study of the convolutional architectures built from k-forms was sadly beyond the scope of the paper, albeit an exciting future direction!
> >
> >     In our graph benchmarks, we use only the non-equivariant features for simplicity, and since we have not yet implemented the full convolution framework and worked out the theory. We have stressed this point in the paper to be sure that this is clear to the reader.  These types of features are things like atomic weight, etc which are clearly not equivariant. Since these features are real-valued, they induce a geometry on the complex, albeit one that is quite different from the picture we have of the molecule when we represent it in $\mathbb{R}^2$ or $\mathbb{R}^3$ as atoms and bonds. In particular, the method works very well on the BACE BBBP and HIV datasets in the MoleculeNet benchmarking set, where the node embeddings are SMILE properties of the nodes - these include both one-hot encoding of atomic types and valence etc. This shows that using feature space embeddings to induce a geometry on the complex can be effectively leveraged using our method.
> > &nbsp;
> >
> > 3. We note that the information of a pre-existing chain can be easily incorporated into the pipeline. Integration of k-simplices in $\mathbb{R}^n$ induces an integral over any k-chain by linearity. In other words, since a k-chain is a linear combination of k-simplices, the integral is the corresponding linear combination of the integrals of each simplex. We explain this formula explicitly in Equation 13, and note that this is a well-established convention for the integration of differential forms over k-chains in algebraic topology and differential geometry. Indeed, the fact that integration produces a singular cochain (Equation 11) implies that every k-chain can be assigned a real number corresponding to its ‘integral’. Another way to think about it is that the real number is the inner product of the dual of the provided chain with the k-form integrated cochain in the standard inner product structure on simplicial cochains. We have modified the paper accordingly.

---

> > ### Comment · Reviewer_gTeC · 2023-11-23
> > **Thank you for your response**
> >
> > Thank you for addressing my comments. Given how close the rebuttal was to the deadline, I did not have time to check the paper again thoroughly, but I have increased my score based on your comments.

---

### Official Review · Reviewer_E4oE · 2023-11-02

**Soundness:** 2 fair
**Presentation:** 2 fair
**Contribution:** 2 fair
**Rating:** 5
**Confidence:** 2

**Summary:**

This paper studies the neural representation learning over simplicial complexes. Specifically, They propose to embed the simplician complex into a integration matrix. The integration matrix is computed by the integral of neural k-forms, which is a linear combination of k-forms weighted by MLP outputs. They argue that many previous works rely on an non-canonical initialisation of feature cochains to compute the representation, while their method bypass this problem as integration matrix does not involve the cochains.

**Strengths:**

- The background introduction is comprehensive and technical part seems sound

**Weaknesses:**

- The baseline GAT, GCN and GIN in experiments are too weak: these are the most basic message passing GNNs. Comparisons with other neural networks on simplicial complexes could make the evaluation more solid.

**Questions:**

- I think the whole pipeline is not very clear to me: when producing the integration matrix using neural k-forms, which are the learnable parameters and which are the hyper-parameters?

---

> ### Author Response · Authors · 2023-11-22
> **Official answer to Reviewer E4oE**
>
> ### Answer weaknesses:
> &nbsp;
>
> Thanks for the remarks concerning the experimental comparison. We added EGNN, a state-of-the-art equivariant GNN, in the revision and observed that, despite only using a fraction of the number of parameters, our method is still competitive and often outperforms it. We did not compare to any simplicial neural networks because these networks are not compatible and applicable to our tests— notice that, for the sake of better comparability, we only use graph classification tasks. Since there is a lack of strong ‘simplicial’ data sets, we found it best to first remain in the realm of graphs.
>
> &nbsp;
>
> ### Answer questions:
>
> &nbsp;
>
> The learnable parameters are the coefficients in the k-forms modelled as MLPs as well as the layers post integration that can act on the integration matrices. The hyper-parameters are the number of hidden layers and the number of cochains to be learned. The model is composed of multiple steps that come together in the following way: The neural $k$-form as an MLP provides a function that can be evaluated point-wise on the embedding space of the graph, however point-wise evaluation of the output is not interesting as it does not tell us anything about the graph or simplicial complex structure. Thus through integration of the k-form on the region of integration determined by the $k$-simplices (a deterministic process without learnable parameters), we can transform the point-wise information into cochains, yielding the integration matrix. The integration matrix can then be processed by further layers that will provide the output prediction. This prediction is used to update the model. The integration part can be understood as an operation that “aggregates” the information of the neural k-form with respect to the structure of the graph.
>
> Given the concerns you raised we have taken two steps to ensure that the details are clear to the reader. Firstly, we have provided a detailed subsection regarding the general architecture and parameters in Section 3. Secondly, we have expanded the explanation of experiments in the body of the paper which we believe gives useful examples of how the basic architecture can be implemented in practice.

---

### Official Review · Reviewer_pNcs · 2023-11-04

**Soundness:** 3 good
**Presentation:** 3 good
**Contribution:** 4 excellent
**Rating:** 6
**Confidence:** 2

**Summary:**

This paper proposes a geometric deep learning method termed neural k-forms. The proposed method leverages geometric information of data and creates representations of simplices using differential k-forms. The authors show the universal approximation property of the neural k-forms in approximating k-forms by applying differential geometry tools. Through several stylized tasks, the authors demonstrate the effectiveness and interpretability of the approach. The proposed neural k-form also outperforms message passing based state-of-the-art graph neural networks in several real-world graph classification benchmark datasets.

**Strengths:**

1. The paper provides an interesting view of MLPs as neural 0-forms and extend it to neural k-forms to harness the geometric information in data via embedded simplicial complexes.
2. The integration of neural forms is introduced to construct finitely-parametrizable, learnable singular cochains such that the neural k-form is usable in practice. Moreover, the integration matrix induced by a neural k-form produces feature cochains that facilitate training process and permit interpretation.
3. Albeit rather technical and math-heavy, the presentation is self-contained and consistent, and the authors provide strong connections between mathematical machinery and model intuition.
4. The proposed method generates interpretable features in synthetic tasks and outperforms standard GNNs by a large margin in several real-world graphs.

**Weaknesses:**

1. The neural k-form relies heavily on the geometric information of the node features and makes use of the simplicial complex structure of data but to a lesser extent the graph information. Learning on graphs with weak or no geometric information can pose a great challenge to the proposed method.
2. The computation feasibility for higher order k-forms and the numerical integration may hamper their application in practice.
3. The approach triumphs in graph classification thanks to the simplicial representations but may suffer in node prediction when smoothing is a blessing.

**Questions:**

1. How does the neural k-form compare to other simplicial neural networks in terms of performance and computation?
2. Is there a way to incorporate convolution in neural k-forms?

Typo: in eq. (6), $d_{x_{i_k}}$ should be $dx_{i_k}$.

---

> ### Author Response · Authors · 2023-11-22
> **Official answer to Reviewer pNcs**
>
> We thank the reviewer for the helpful comments about our manuscript. We have made some changes according to these comments. A detailed answer to the weaknesses and questions can be found below as well as a description of the changes made to the paper accordingly. In the manuscript the main changes have been added in blue for more clarity.
> &nbsp;
>
>
> ### Answer weaknesses:
> &nbsp;
>
> 1. After the helpful comments from yourself and the other reviewers, we realise that we could have done a better job of stressing how the graph/simplicial structure determines its vector representation. The graph information is incorporated as follows: the nodes are embedded in the ambient feature space and the edges (1-simplices) serve as regions of integration for 1-forms. Thus, if the edge structure changes, the 1-forms will be integrated over a different region, and produce a different vector representation of the graph. For example, two graphs with the same embedded nodes but different edges can have dramatically different integrals against the same form.
>
> It is true that if the node embeddings are relatively uninteresting geometrically - for example, one hot encoding of atom types etc. - the simplicial regions of integration are a bit less intuitive. However, the integral can still be calculated in the same manner, and despite the ‘discreteness’ of the node embeddings the method performs well. When moving to higher-dimensional, real-valued scalar node features the method indeed performs much better, and overall we agree with your intuition that real-valued, node embeddings are preferable as inputs for our algorithm. &nbsp;
>
>
> 2. Indeed the computational cost of integrating $k$-forms with $k>2$ becomes quite expensive. Given the results in this paper we believe that an important next step to this paper would be to better develop the computational aspect of this method in higher dimensions. One possible trick that we could use is the isomorphism between 2-forms and 1-forms in $\mathbb{R}^3$, which in theory should speed up computing surface integrals for surfaces in $\mathbb{R}^3$ considerably. We plan to investigate this in future work. &nbsp;
>
> 3. For the moment, we have decided to focus on classification only due to space constraints in the paper. However, the method could easily be adapted for node prediction and we put this as future work. For example, one could integrate a set of 1-forms across the local 1-neighbourhood of each node to produce a vectorization that could serve as basis for node-based learning tasks. &nbsp;
>
>
>
> ### Answers to questions &nbsp;
>
> 1. We opted to restrict benchmarking to graph classification, since it has a more established benchmarking framework. This was the approach taken by one of the main papers on simplicial neural networks, Bodnar et. al [1]. We did not directly compare to this framework, but based on the reported results in that paper it is likely our algorithm is competitive with them. &nbsp;
>
> [1]  Bodnar, C., Frasca, F., Wang, Y., Otter, N., Montufar, G.F., Lió, P. &, Bronstein, M.. (2021). Weisfeiler and Lehman Go Topological: Message Passing Simplicial Networks. Proceedings of the 38th International Conference on Machine Learning, in Proceedings of Machine Learning Research 139:1026-1037 Available from https://proceedings.mlr.press/v139/bodnar21a.html. &nbsp;
>
>
> 2. Yes! This is entirely feasible and in our view an exciting future direction. Due to questions raised by this reviewer and others we have decided to include an example of convolutional 1-form network in the *Additional Experiments* section in the appendix and how this addresses the reviewer’s question.

---

> ### Author Response · Authors · 2023-11-23
>
> We appreciate your positive assessment of your paper. May we kindly ask what prompted the change in your score from an initial 8 to a 6? We believe that we not only addressed all reviewer concerns but also that our paper benefitted from these changes with respect to clarity.
>
> Since the deadline for the rebuttal is coming up soon, please let us know if there's anything else we can do to address your concerns. Such a score change without any comments puts us at a fundamental numerical disadvantage with respect to more applied papers. We would love to answer any questions you might have!

---

### Official Review · Reviewer_VjaY · 2023-11-09

**Soundness:** 3 good
**Presentation:** 3 good
**Contribution:** 2 fair
**Rating:** 5
**Confidence:** 4

**Summary:**

In this paper, the authors introduced a new method, neural k-form,  of utilizing integration of differential forms over embedded simplicial complexes for robust and interpretable data representations. The authors provided necessary background information. Then they introduced the basic idea of neural k-form, and showed that neural k-form can approximate any k-form with compact support. The authors discussed the concept of integration matrices and their basic properties and presented numerical examples of classification problems utilizing the neural k-form method.

**Strengths:**

The authors tap into the rich area of topology and geometry, established a potentially very powerful connection.

**Weaknesses:**

Overall, I feel that while the method proposed in the paper has potential it has not been fully developed and analyzed.

Many statements, for example, "Thus an l-tuple of k-forms produces an l-dimensional representation of the simplex independently of any message passing" in the introduction and "In practice, this will facilitate the training process and permit interpreting the learned cochains" in remark 5, are not backed up by detailed analysis.

The neural k-form learning architecture is presented very briefly in the caption of a figure, it is very difficult for others to figure out how to implement the method.

It is understandable audience would want to find out the performance of the method proposed for high dimensional problems. Unfortunately, the experiments are limited to 1-form and 2-form, and the differences in performance are not very significant except in the last experiments.

**Questions:**

On page 3, why is the parallelepiped spanned by tangent vectors called "infinitesimal parallelepiped"? The intuition and the formal definition are not very well connected.

On page 4, second paragraph, should $b\in R^{1\times m}$ instead of $b\in R^{1\times n}$?

Can you provide an argument for your stamens on the dimension of singular k-chains?

On Page 15, right below equation (38), should it be $\alpha_{i,j}$ instead of $f_{i,j}$?

---

> ### Author Response · Authors · 2023-11-22
> **Official Answer to reviewer  VjaY - part 1**
>
> We thank the reviewer for the helpful comments and the careful reading of our manuscript. Similar comments about the architecture of the method have been made by other reviewers. We have addressed these comments in the new version of the manuscript. The main changes have been incorporated in blue for an easier overview. Below we answer the specific comments and questions detailing the changes we have made.
> &nbsp;
>
>
> ### Answer to weaknesses:
>
> 1. The fact that our algorithm produces a representation without message-passing is a basic property of the architecture. This seems to have been lost in our presentation of the paper, since many of the reviewers seemed confused about this point. We have added a new section called *Architecture and Parameters* to clarify how the procedure works in practice.
>
>     Simplicial message-passing (see Bodnar et al. [1]) layers take a matrix of cochains as input, and produce another matrix of cochains as output before readout and downstream analysis. The representations of each simplex correspond to rows of this matrix when it is in the standard basis of simplicial chains. The point is that our algorithm replaces message passing by generating a matrix of cochains via integration matrices independently of any message passing. We have put this in the paper just above Remark 8 at the end of the new section to clarify, and hope that it addresses the reviewer’s concerns.
> &nbsp;
> We agree with the reviewers comment regarding the last line in Remark 5 and have removed it from the paper.
>
> [1]  Bodnar, C., Frasca, F., Wang, Y., Otter, N., Montufar, G.F., Lió, P. & Bronstein, M.. (2021). Weisfeiler and Lehman Go Topological: Message Passing Simplicial Networks. Proceedings of the 38th International Conference on Machine Learning, in Proceedings of Machine Learning Research 139:1026-1037 Available from https://proceedings.mlr.press/v139/bodnar21a.html.
> &nbsp;
> &nbsp;
>
>
> 2. Again, since many reviewers found it difficult to understand the general architecture, we have expanded on the figure by adding a whole sub-section *Architecture and Parameters* in Section 3.
>
>
> 3. It is slightly unclear whether the reviewer is referring to the dimensionality of the node embedding or the dimensionality of the form. For high dimensional node embeddings, 1-form computations are still efficient as shown in our real-world graph case.
>
>     We understand the general concerns about showing higher dimensional form examples. We have decided to benchmark the method on well-known graph data sets as is usually done in the graph learning community. We  show an example as a proof of concept with $2$-forms and $2$-dimensional complexes, to show how the method works in the higher dimensional case. Unfortunately data sets with higher dimensional structure are not widely available and rarely used as benchmarks hence the decision of having only an synthetic example for the higher dimensional case. In addition we believe that the difficult step in understanding the theory behind our model was going from one dimension to any dimensions higher than 1. Understanding and visualization of 2-dimensional objects being easier to communicate about, we decided to keep the proof of concept to two dimensions only, as a way to show that one can go from simple 1-dimensional case  to more complicated multidimensional cases.
>
>     About the performance, we believe that the strength of our method is due to its geometric foundations. In the experiments shown in Table 1 the point is not only that our model performs better than or similar to other models, but also that the number of parameters needed is smaller, this aspect is even more prevalent in the second set of experiments (Table 2) where not only the number of parameters is significantly smaller than other standard models but our method performs the tasks much better.
> &nbsp;

---

> > ### Author Response · Authors · 2023-11-22
> > **Official Answer to reviewer VjaY - part 2**
> >
> > ### Answer to questions:
> >
> > 1. We have removed the terminology ‘infinitesimal parallelepiped’. In its place, we have tried to describe how differential forms generalise the properties of the determinant for intuition. In general, we feel that it is difficult to give an adequate feeling for the differential forms in the space constraints and that the metaphor of  ‘infinitesimal parallelepiped volumes’ is probably more confusing than helpful.
> >
> >     To still provide a bit of intuition, we instead (1) added the example of the correspondence between vector field and 1-form to the body of the paper which we hope serves as an example for the reader to hold onto throughout the text and (2) commented on how alternating multi-linearity generalises the determinant. We hope that the updated version is enough to give the reader the idea, and that the detailed Appendix A is sufficiently self-contained for the motivated reader.
> >
> >
> > 1. The MLP definition which contained this typo has been removed from the revised version.
> >
> >
> > 1. The dimension of singular $k$-chains has a basis given by all possible continuous maps from the $k$-chains into the $\mathbb{R}^n$ - this is clearly infinite, and not countably. Even the singular 0-simplices, the basis of which is bijective correspondence with points in $\mathbb{R}^n$, is an uncountably generated real vector space. We have mentioned this as a footnote to avoid confusion.
> >
> >
> > 1. Thank you for pointing out this typo, it has been corrected.

---

### Author Response · Authors · 2023-11-22

We thank all of their reviewers for their constructive reviews! There were several general overall trends in the reviews which we thought warranted a general comment. All the changes in the revision we added in blue to aid the reviewers.

Firstly, the main point was that the actual architecture was unclear to the reviewers. We have added a new section called *Architecture and Parameters* to clarify how one would go about implementing our algorithm. We have also tried to address individual questions in our responses to reviewers and used this to guide the writing of the new section in the revised edition. We found the space by compressing the read-out layer section and some general pruning of the more mathematical sections.

Secondly, the reviewers broadly agreed that not enough information about the experiments was given. We have greatly expanded the information about the setup, data, and architecture in each experiment. We found the space for this by omitting the weakest experiment concerning the simplicial 1-Laplacian eigenvectors.

Finally, some reviewers raised concerns about how our architecture deals with equivariances in the ambient embedding. We have clarified that, for now, our architecture ignores this type of information. We have added a small example of a convolutional 1-form network in the appendix which gives a prospective direction for dealing with such types of equivariances.

---

### Meta-Review · Area_Chair_Cxtj · 2023-12-12

**Metareview:**

The paper proposes an alternative approach to *message passing* method used in Geometric Deep Learning. The main idea is to leverage a duality between MLP and differential form. The input object is a collection of simplex, and the differential form serves as a feature map for them. The authors argue that this capture geometrical semantics and learning is still computational feasible since differential form can be integrated. An adaptation of the universal approximation theorem, shows that k-forms with compact support can be uniformly approximated by neural k-form with one hidden layer. Finally some numerical experiments show encouraging performances.


> Presentation

Overall, the paper is difficult to read. If one is already familiar with differential form, one will hardly understand the ML part, which is only roughly explained (readout layers, architecture and parameters are actually very short and not really informative). If one is not familiar with differential form, it is quite ambitious to expect to understand the form from such a dense, math heavy background recall, that are most of the time not even used.

The paper should focus more on explaining the motivation behind the current approach and how it overcomes previous limitations of the message passing approach. So additional details on message passing and its limitations would be beneficial in order to appreciate the contributions.

Also as expected, the universal approximation theorem is nowhere used and offer limited insight. At least the authors did not provide enough discussions on that.


> Experiments

The benchmarks in Table 1 do not show better results compared to existing methods. However the table 2 display quite impressive improvement while using a smaller number of parameters.  This are results on a *single* dataset.


> Minor:

The expression "non polynomial activation functions" is not very clear.

It would be nice to have examples when the abstract concept is introduced. For example in the section *Cochains and representation Learning" It is might be good idea to concretely relate these concepts to practical machine learning in order to improve readability.


On the computational side, integration in high dimension is still not that easy but the authors did not really comments on how it impacts the proposed methods.

**Justification For Why Not Higher Score:**

The numerical experiments are quite limited and the presentation of the method must be significantly improved.

**Justification For Why Not Lower Score:**

The alternative approach to message passing is original and one might expect nice follow up paper. The overall scores indicate that the reviewers are leaning toward (borderline) acceptance

---

### Decision · Program_Chairs · 2024-01-16

Accept (poster)